# PNPLA3 is a triglyceride lipase that mobilizes polyunsaturated fatty acids to facilitate hepatic secretion of large-sized very low-density lipoprotein

Scott M. Johnson [1,2,3], Hanmei Bao[4], Cailin E. McMahon[5], Yongbin Chen [1], Stephanie D. Burr [1], Aaron M. Anderson [6], Katja Madeyski-Bengtson[7], Daniel Lindén [8,9], Xianlin Han [4] & Jun Liu [1,10] ✉

The I148M variant of PNPLA3 is closely associated with hepatic steatosis. Recent evidence indicates that the I148M mutant functions as an inhibitor of PNPLA2/ATGL-mediated lipolysis, leaving the role of wild-type PNPLA3 undefined. Despite showing a triglyceride hydrolase activity in vitro, PNPLA3 has yet to be established as a lipase in vivo. Here, we show that PNPLA3 preferentially hydrolyzes polyunsaturated triglycerides, mobilizing polyunsaturated fatty acids for phospholipid desaturation and enhancing hepatic secretion of triglyceride-rich lipoproteins. Under lipogenic conditions, mice with liver-specific knockout or acute knockdown of PNPLA3 exhibit aggravated liver steatosis and reduced plasma VLDL-triglyceride levels. Similarly, I148M-knockin mice show decreased hepatic triglyceride secretion during lipogenic stimulation. Our results highlight a specific context whereby the wild-type PNPLA3 facilitates the balance between hepatic triglyceride storage and secretion, and suggest the potential contribution of a loss-of-function by the I148M variant to the development of fatty liver disease in humans.

Metabolic dysfunction-associated fatty liver disease (MAFLD) is presently the most prevalent form of chronic liver disease unrelated to alcohol consumption[1–3]. The development of MAFLD occurs when hepatic steatosis, the accumulation of triglycerides (TGs) in hepatocytes, becomes persistent due to disruptions in the balance between metabolic pathways responsible for TG synthesis, hydrolysis, and secretion[4–8]. Hepatic TG secretion is facilitated by very-low density lipoproteins (VLDLs); disturbances in VLDL secretion not only leads to the buildup of TGs but also facilitates the accumulation of lipotoxic lipids carried by VLDLs, thus contributing to the progression of the disease[9]. Notably, compared to simple steatosis, metabolic dysfunction-associated steatohepatitis (MASH) is characterized by

[1]Department of Biochemistry and Molecular Biology; Mayo Clinic College of Medicine & Science, Rochester, MN 55905, USA. [2]Mayo Clinic Graduate School of Biomedical Sciences; Mayo Clinic College of Medicine & Science, Rochester, MN 55905, USA. [3]Department of Cell Biology; University of Texas Southwestern Medical Center, Dallas, TX 75390, USA. [4]Barshop Institute for Longevity and Aging Studies and Department of Medicine, Division of Diabetes; University of Texas Health San Antonio, San Antonio, TX 78229, USA. [5]Molecular Biology and Genetics Department; Cornell College of Agriculture and Life Sciences, Ithaca, NY 14853, USA. [6]Department of Developmental Biology; Washington University School of Medicine in St. Louis, St. Louis, MO 63110, USA. [7]Translational Genomics, Discovery Sciences; BioPharmaceuticals R&D, AstraZeneca, Gothenburg, Sweden. [8]Bioscience Metabolism, Research and Early Development Cardiovascular, Renal and Metabolism (CVRM); BioPharmaceuticals R&D, AstraZeneca, Gothenburg, Sweden. [9]Division of Endocrinology, Department of Neuroscience and Physiology; Sahlgrenska Academy, University of Gothenburg, Gothenburg, Sweden. [10]Division of Endocrinology, Diabetes, Metabolism and Nutrition; Mayo Clinic in Rochester, Rochester, MN 55905, USA. ✉e-mail: liu.jun@mayo.edu

reduced overall secretion of VLDL and impaired production of apolipoprotein B (ApoB)[10–13]. Conversely, the excessive storage of lipids in MAFLD stimulates the secretion of VLDL[14–16], thereby promoting dyslipidemia.

The assembly of VLDL in hepatocytes initiates in the endoplasmic reticulum (ER) membrane and progresses within the ER lumen and Golgi apparatus. ApoB serves as the primary structural component of VLDL, binding both neutral lipids and phospholipids (PLs). The assembly of ApoB-containing lipoproteins occurs in a two-step process, commencing in the ER lumen with the lipidation and stabilization of ApoB by the microsomal triglyceride transfer protein (MTTP)[17–19]. As the ApoB lipoproteins grow, they acquire increasing amounts of TG along with major PLs such as phosphatidylcholine (PC), phosphatidylethanolamine (PE) and sphingomyelin (SM) to expand their surface area[18,20]. Additionally, maintaining the appropriate content of polyunsaturated fatty acids (PUFAs) in ER PLs appears necessary for effective transfer of TGs across the membrane for VLDL assembly[21–24]. Studies in mice lacking hepatic Lpcat3, a critical ER enzyme regulating PUFA abundance (e.g., linoleate and arachidonate) in PCs, demonstrate reduced plasma TG levels, impaired secretion of large-sized VLDL, and increased hepatic steatosis[21]. In addition, impaired hepatic lipid synthesis from PUFAs was shown to contribute to PC deficiency and hepatic steatosis in TM6SF2 E167K variant carriers[24].

Under normal fasting conditions, FAs used for hepatic VLDL-TG synthesis are primarily derived from adipose tissue lipolysis. However, in MAFLD, de novo lipogenesis (DNL) in the liver is significantly increased, with over 25% of VLDL-TG stemming from DNL[25]. Although both promote hepatic steatosis, lipogenic high-sucrose diets (HSD) enhance VLDL secretion and suppress hepatic FA oxidation, while high-fat diets (HFD) often exert the opposite effects[26]. In response to dietary sugar, increased glucose metabolism and elevated insulin signaling stimulate the activity of lipogenic transcription factors Carbohydrate response element binding protein (ChREBP) and Sterol regulatory element-binding protein 1c (SREBP-1c), which together mediate the expression of enzymes involved in FA and TG synthesis[27–29]. In addition, activation of another transcription factor liver X receptor (LXR) by cellular oxysterols or pharmacological agonists can also activate SREBP-1c, facilitating DNL and hepatic TG secretion[30,31]. SREBP-1 activation is associated with increased VLDL-TG synthesis[32,33], whereas ChREBP facilitates the expression of crucial proteins involved in VLDL lipidation and assembly (MTTP, TM6SF2)[34]. These observations underscore the strong link between DNL, VLDL synthesis and secretion, and dyslipidemia. Since ω3 and ω6 desaturases are absent in mammals, de novo FA synthesis predominantly yields saturated (SFAs) and monounsaturated FAs (MUFAs). Thus, a key question arises as to how the PUFA content in PLs is sustained in support of VLDL synthesis during lipogenic stimulation.

A common genetic variant of patatin-like phospholipase domain-containing protein 3 (PNPLA3) (rs738409), which encodes an I148M substitution, is strongly associated with an increased susceptibility to MAFLD in humans[35]. PNPLA3 is a close paralog of PNPLA2/ATGL[36], the key intracellular triglyceride lipase[37–43]. The TG hydrolase activity of ATGL is robustly activated by comparative gene identification 58 (CGI-58)[44], and potently inhibited by G0/G1 switch gene 2 (G0S2)[45]. Early biochemical studies of PNPLA3 suggested TG hydrolase and acyltransferase activities[39,46]. However, subsequent studies aimed to recapture these enzyme activities have yielded inconsistent results. In a well-established in vitro lipase assay, PNPLA3 showed insignificant hydrolytic activity toward triolein substrate when compared to ATGL[47]. In mice, overexpression of the wild-type PNPLA3 in liver failed to impact hepatic TG content or diet-induced liver steatosis[48,49]. More intriguingly, when the PNPLA3 protein was knocked out, mice displayed no significant metabolic phenotypes under the conditions of either high-fat or high-sucrose feeding[50,51]. As such, much effort has been devoted to uncovering the association between the I148M mutant and disease, leaving unaddressed the physiological relevance of the wild-type PNPLA3 in the regulation of hepatic lipid homeostasis.

Recent studies have demonstrated that the I148M mutant of PNPLA3 exhibits greater protein stability and a higher tendency to accumulate on lipid droplets (LDs)[49,52,53]. The mutant protein was shown to suppress ATGL-mediated lipolysis by competing for its coactivator CGI-58 at the LD surface[54]. In this regard, the function of PNPLA3 has not been evaluated in an ATGL-independent context. Moreover, while these data generally support an ATGL-dependent gain-of-function paradigm, evidence obtained from human studies has suggested the possibility of PNPLA3-I148M variant being a loss-of-function allele. Based on the association of I148M variant with an increased enrichment of PUFA in hepatic TGs and a decreased PUFA content in VLDL-TGs, it was proposed that the wild-type PNPLA3 functions as a diacylglycerol (DAG) transacylase or hydrolase to specifically transfer PUFA for PL synthesis[55]. However, despite displaying higher liver fat content, earlier work on VLDL secretion in human homozygous carriers of the I148M variant has yielded inconsistent results[56,57]. Therefore, the precise role PNPLA3 plays in the hepatic VLDL synthesis and secretion remains undetermined. To our knowledge, studies conducted thus far have not attempted to distinguish the dietary circumstances in which the I148M variant could affect plasma VLDL and lipid profiles in humans.

Although it is abundantly expressed in adipose tissue and liver like ATGL, the expression of PNPLA3 is upregulated by feeding instead of fasting[58,59]. In the liver, carbohydrate feeding triggers increased PNPLA3 expression through the LXR/SREBP-1c pathway[60], suggesting PNPLA3 functions under lipogenic rather than lipolytic conditions. In this regard, the present study set out to achieve two major objectives: to elucidate the lipolytic function of PNPLA3 independent of ATGL and uncover its physiological relevance during lipogenic stimulation. To that end, we utilized ATGL-deficient cell lines and mice for gain-of-function experiments. For loss-of-function analyses, we generated a liver-specific PNPLA3 knockout mouse model. We also employed a previously validated antisense oligonucleotide (ASO)[61] to knock down PNPLA3 in primary mouse hepatocytes and adult mice. Additionally, we utilized knock-in (KI) mice to assess the impact of the I148M mutation on hepatic TG secretion. Here, we demonstrate that PNPLA3 possesses a selective lipase activity for PUFA-containing TGs and is critically involved in the secretion of large-sized VLDLs during lipogenic stimulation. Our findings highlight the physiological role of wild-type PNPLA3 in vivo, and shed light on the specific context in which PNPLA3 is essential for balancing hepatic and circulating lipid levels. We also provide evidence suggesting that the loss of such a function by the I148M variant may contribute to hepatic steatosis in human carriers.

## Results

### PNPLA3 degrades PUFA-LDs in an ATGL-independent manner

To test the ATGL-independent role of PNPLA3 in intracellular lipolysis, we ectopically expressed PNPLA3 fused at the N-terminus with a FLAG epitope tag (FLAG-PNPLA3) in ATGL[−/−] HeLa cells that were generated previously[62]. To enrich intracellular TG-LDs with either monounsaturated FAs (MUFA) or various types of PUFAs, cells were treated overnight with 18:1 n-9 oleic acid (OA), 18:2 n-6 linoleic acid (LA), 18:3 n-3 α-linolenic acid (αLA), 20:4 n-6 arachidonic acid (AA), or a mixture of 22:6 n-3 DHA and 20:5 n-3 EPA. While LDs were stained with the neutral lipid dye BODIPY 493/503, FLAG-PNPLA3 was detected by immunofluorescence staining. As shown in Fig. 1a, b, OA treatment induced LD accumulation to similar levels in PNPLA3-expressing cells and adjacent untransfected cells. In contrast, when cells were treated with LA, PNPLA3 expression resulted in an almost complete loss of LD accumulation. Interestingly, PNPLA3-expressing HeLa cells treated with other PUFAs also failed to accumulate LDs. The LD-reducing effect

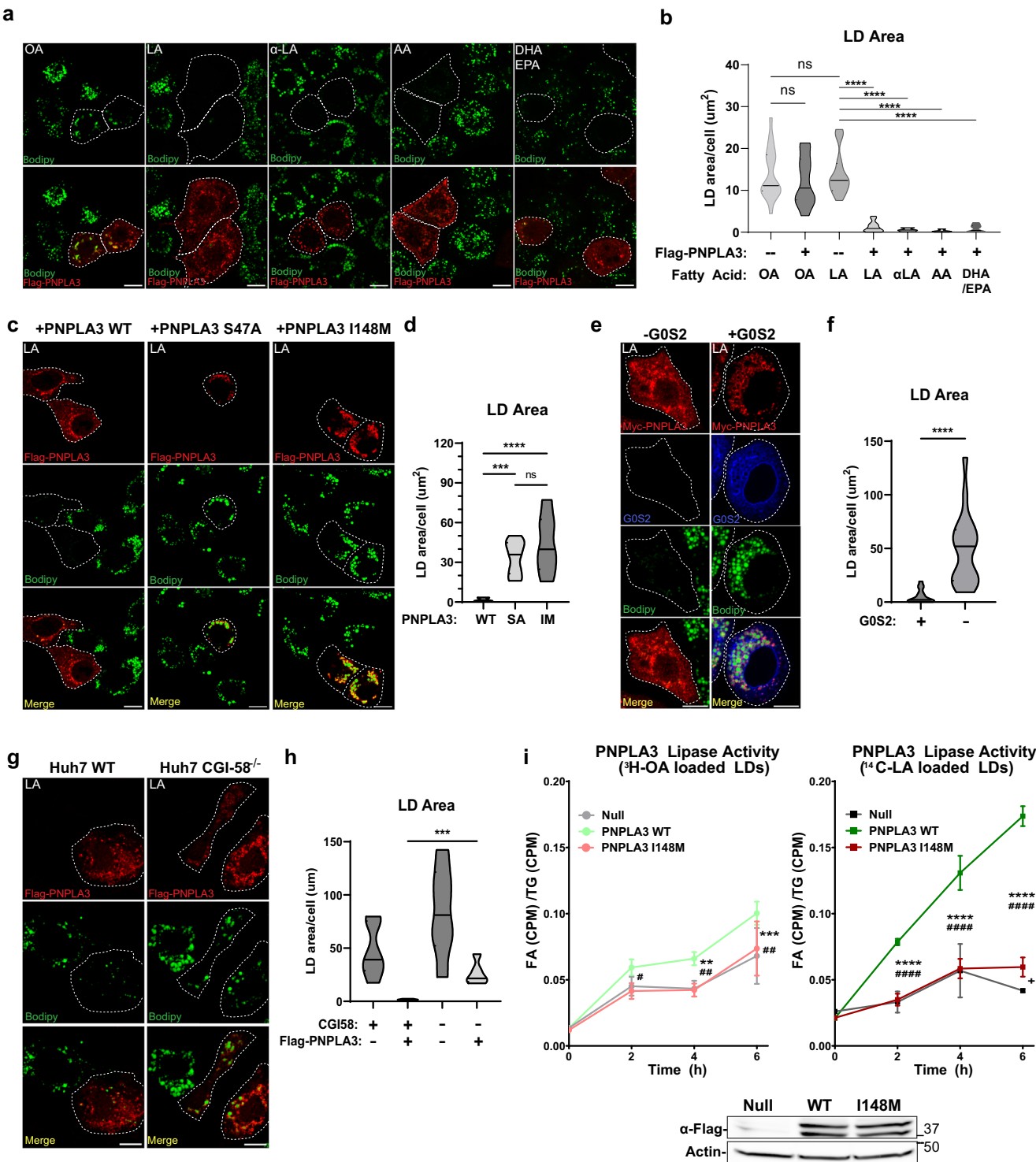

of PNPLA3 was abolished when either the catalytic serine in its patatin-like hydrolase domain was mutated to alanine or the clinically relevant I148M mutation was introduced (Fig. 1c, d). Likewise in mouse primary hepatocytes, LA treatment produced considerably fewer LDs than OA treatment when PNPLA3 was adenovirally overexpressed, an effect lost by expression of the I148M mutant (Supplementary Fig. 1a–d). Of note, the LD clearance was less complete in primary hepatocytes likely due to the number of LDs with variable composition that existed prior to FA treatment.

In ATGL$^{-/-}$ HeLa cells, coexpression of ATGL inhibitor G0S2 was sufficient to block LD degradation by PNPLA3 (Fig. 1e, f). Moreover,

loss of CGI-58 significantly impaired the ability of FLAG-PNPLA3 to degrade LA-induced LDs when CGI-58$^{-/-}$ Huh7 cells were compared to the wild-type control cells (Fig. 1g, h).

To assess PNPLA3 lipolytic activity directly we employed a standard lipase assay with slight modification. Traditional approaches to assay lipase activity entail the incubation of purified enzyme or cell lysates with radiolabeled TG substrates emulsified by phospholipids[41,45]. However, incubation of PNPLA3 expressing cell lysates with trilinolein (54:6) showed minimal FA release. We hypothesized that our inability to capture PNPLA3 lipase activity in vitro is likely due to the inability of extracted PNPLA3 to properly localize to

**Fig. 1 | PNPLA3 degrades PUFA-LDs independently of ATGL. a** ATGL$^{-/-}$ HeLa cells transfected with FLAG-PNPLA3 were treated overnight with (from left to right) 250 μM of OA, LA, α-LA, AA (250 μM), or DHA (125 μM) + EPA (125 μM), and then immunostained with a FLAG antibody. LDs were stained with BODIPY. **b** Quantification of LD area in (**a**). **c** ATGL$^{-/-}$ HeLa cells transfected with FLAG-tagged (from left to right) PNPLA3-WT, PNPLA3-S47A, or PNPLA3-I148M were treated overnight with 250 μM LA and then imaged as in (**a**). **d** Quantification of LD area in (**c**). **e** ATGL$^{-/-}$ HeLa cells transfected with Myc-tagged PNPLA3 or co-transfected with Flag-G0S2 were treated overnight with 250 μM LA and then imaged as in (**a**). **f** Quantification of LD area in (**E**). **g** WT control (left) or CGI-58$^{-/-}$ (right) Huh7 cells transfected with FLAG-PNPLA3 were treated with 250 μM LA overnight and then imaged as in (**a**). **h** Quantification of LD area in (**g**). **i** PNPLA3$^{-/-}$ primary mouse hepatocytes were isolated and infected with null, PNPLA3 WT or PNPLA3 I148M expressing adenovirus, and loaded with 250 μM OA + $^3$H-OA (left) or 250 μM LA + $^{14}$C-LA (right). After cell lysis, isolated LDs were normalized for total TG content and incubated at 37 °C with 20 μM ATGListatin in a 5% FA-free BSA solution. At the indicated intervals, fractional aliquots were removed from the reaction mixture, and albumin-bound FA was extracted for scintillation counting (*n* = 4 per condition from a representative experiment). Scale bar in microscopic images = 10 μm. Error

bars in (**i**) represent mean ± SD. (*) indicates statistical significance between [Null] and [PNPLA3 WT], (#) indicates statistical significance between [PNPLA3 WT] and [PNPLA3 I148M], and (+) indicates statistical significance between [Null] and [PNPLA3 I148M]. Box-plot elements: center line, median; box limits, upper and lower quartiles. *$p$ < 0.05, **$P$ < 0.01, ***$P$ < 0.001, ****$P$ < 0.0001, two-sided *t* test. Specific *p* values: **b** Untransfected+OA vs PNPLA3 + OA: *p* = 0.488; Untransfected +OA vs Untransfected+LA: *p* = 0.961; Untransfected+LA vs PNPLA3 + LA: *p* = 1.22E-08; Untransfected+LA vs PNPLA3+aLA: *p* = 1.35E-06; Untransfected+LA vs PNPLA3 + AA: *p* = 2.25E-08; Untransfected+LA vs PNPLA3 + DH/EPA: *p* = 1.59E-08. **d** PNPLA3 WT + LA vs PNPLA3 S47A + LA: *p* = 2.46E-05; PNPLA3 WT + LA vs PNPLA3 I148M + LA: *p* = 6.43E-04; PNPLA3 S47A + LA vs PNPLA3 I148M + LA: *p* = 0.237. **f** *p* = 9.60E-06. **h** *p* = 9.22E-04. **i** OA loaded LDs: Null vs PNPLA3 WT/2 h: *p* = 0.0235, 4 h: *p* = 0.0011, 6 h: *p* = 0.030; Null vs PNPLA3 I148M/2 h: *p* = 0.458, 4 h: *p* = 0.804, 6 h: *p* = 0.715; PNPLA3 WT vs PNPLA3 I148M/2 h: *p* = 0.0060, 4 h: *p* = 0.00051, 6 h: *p* = 0.0538; LA loaded LDs: Null vs PNPLA3 WT/2 h: *p* = 3.63E-05, 4 h: *p* = 8.38E-04, 6 h: *p* = 3.91E-08; Null vs PNPLA3 I148M/2 h: *p* = 0.708, 4 h: *p* = 0.885, 6 h: *p* = 0.0029; PNPLA3 WT vs PNPLA3 I148M/2 h: *p* = 2.68E-06, 4 h: *p* = 6.87E-05, 6 h: *p* = 6.02E-07.

---

the surface of emulsified TG substrates. On this basis we modified our experimental approach, utilizing primary hepatocytes to concurrently express PNPLA3 and produce radiolabeled TG-LDs to ensure the association of the enzyme with LDs (see Materials and Methods). Upon this modification, WT PNPLA3 showed substantially more FA release from LA-labeled LDs than from OA-labeled LDs (Fig. 1i). As in our cell model, this substrate-specific lipase activity was significantly impaired with the presence of the I148M mutation. Taken together, these results indicate that PNPLA3 specifically degrades PUFA-containing LDs through ATGL-independent TG hydrolysis, a function that appears to be repressed by G0S2, enhanced by CGI-58, and lost by the I148M mutant.

### PNPLA3 abrogates hepatic steatosis and mobilizes PUFAs from TGs to PLs in ATGL$^{-/-}$ liver

To gain a deeper insight into the ATGL-independent effect of PNPLA3 on hepatic lipid metabolism in vivo, we overexpressed FLAG-PNPLA3 in the liver of wild-type and ATGL knockout (ATGL$^{-/-}$) mice via adenovirus-mediated gene transfer. Compared with Ad-null, Ad-FLAG-PNPLA3 produced comparable levels of FLAG-PNPLA3 expression in the livers of wild-type and ATGL$^{-/-}$ mice (Fig. 2a). While it elicited no effect on hepatic TG content in the wild-type mice, FLAG-PNPLA3 expression fully ameliorated the hepatic TG accumulation associated with the loss of ATGL (Fig. 2b). The plasma TG levels, however, trended higher in the ATGL$^{-/-}$ mice expressing FLAG-PNPLA3 (*p* = 0.0787) (Fig. 2c). Lipidomic analysis revealed that loss of ATGL increased the abundance of all TG species in the liver, especially those with shorter acyl chains and fewer double bonds (Fig. 2d). Interestingly, PNPLA3 expression led to significant decreases in the abundance of liver TGs with longer acyl chains and more double bonds, an effect that was more drastic in the ATGL$^{-/-}$ mice than in the wild-type mice (Fig. 2d). Consistent with the effects on overall liver TG levels, PNPLA3 expression reduced the absolute abundance of individual TG-FAs, including 16:0, 18:2, 18:1, and 18:0, in the ATGL$^{-/-}$ mice to levels seen in the wild-type animals (Fig. 2e). Compositionally, the fraction of 18:2 and 18:3 were particularly reduced in the liver TGs of ATGL$^{-/-}$ mice, while 18:0 exhibited a profound enrichment (Fig. 2f). Moreover, PNPLA3 did not affect either total PL amount or FA composition of PLs in the liver of wild-type mice (Fig. 2g, h). In comparison, ATGL$^{-/-}$ mice expressing FLAG-PNPLA3 saw elevated levels of total PLs due to significant increases of PC and PE (Fig. 2g, h). Of note, the most enriched PE species were those containing FAs 16:0 and 22:6, while the most enriched PCs contained 16:0 and 18:2 (Fig. 2i, j). These data suggest that in the absence of ATGL, PNPLA3 is highly effective as a lipase that mobilizes

PUFAs from TGs to PLs in the liver. As SREBP-1c is hyper-activated in the livers of ATGL$^{-/-}$ mice[63], our results further suggest a relevant role for PNPLA3 in the context of lipogenic stimulation.

### Loss of PNPLA3 aggravates hepatic steatosis while lowering plasma TG in response to a PUFA-enriched lipogenic diet

To determine the liver-specific function of PNPLA3 in vivo, we generated PNPLA3-floxed (PNPLA3$^{fl/fl}$) mice (Supplementary Fig. 2a). At 8–10 weeks of age, these mice were injected with AAV-TGB-Cre to induce selective disruption of the PNPLA3 gene in hepatocytes (PNPLA3-LKO) (Supplementary Fig. 2b) (Fig. 3a). RT-qPCR analysis revealed an almost complete loss of PNPLA3 mRNA expression in the liver but not in WAT or BAT of Cre-injected mice (Fig. 3b, Supplementary Fig. 2c, d). Based on the evidence that hepatic PNPLA3 is upregulated by carbohydrate feeding[47,59,60] and functions as a PUFA-TG lipase, we sought to create a dietary context in which liver would be enriched with PUFA-TGs while being maintained in a lipogenic state. To this end, we treated the mice with a corn oil-enriched Western diet (COWD) (Research Diets, D21050712i; Supplementary Table s3) supplemented with high glucose and fructose in the drinking water. The COWD contained low levels of SFA (12.9%) and high levels of PUFA (62%). As in the corn oil, the PUFA in COWD was primarily LA (18:2). Prior to tissue and plasma collection, mice were fasted from solid food for 16 h to eliminate contributions of chylomicrons to plasma TG. To allow for sustained lipogenic stimulation, they retained access to glucose/fructose water (Fig. 3a). Livers from PNPLA3-LKO mice showed more visual discoloration (Fig. 3c) and a pronounced increase in fat vacuoles histologically (Fig. 3d). Compared to AAV-null-treated controls, PNPLA3-LKO mice had significantly more hepatic TG accumulation and a concomitant decrease in plasma TG levels (Fig. 3e, f). In addition, ER stress response and activation of the lipogenic transcription factor SREBP-1c are known to be influenced by ER membrane saturation levels[23,64]. However, these processes did not appear to be affected by PNPLA3 deletion as no differences were observed in hepatic expression of lipogenic genes (Srebp1, ChREBP, FASN, ACC1, SCD1, and ACLY), ER stress markers (XBP1, CHOP, and ATF4), or proteins involved in intracellular lipolysis (ATGL and CGI-58) (Fig. 3g–i). Of note, transcript levels of all lipogenic genes probed were slightly depressed by PNPLA3 knockout, though not statistically significant.

Similar changes in TG metabolism were observed when we acutely knocked down PNPLA3 in wild-type mice (Fig. 3j). Specifically, administration of a previously established ASO[61], which suppressed hepatic PNPLA3 expression by ~80% (Fig. 3k), significantly elevated hepatic TG (Fig. 3l) and diminished plasma TG levels (Fig. 3m) under the same dietary conditions when compared to a scrambled ASO

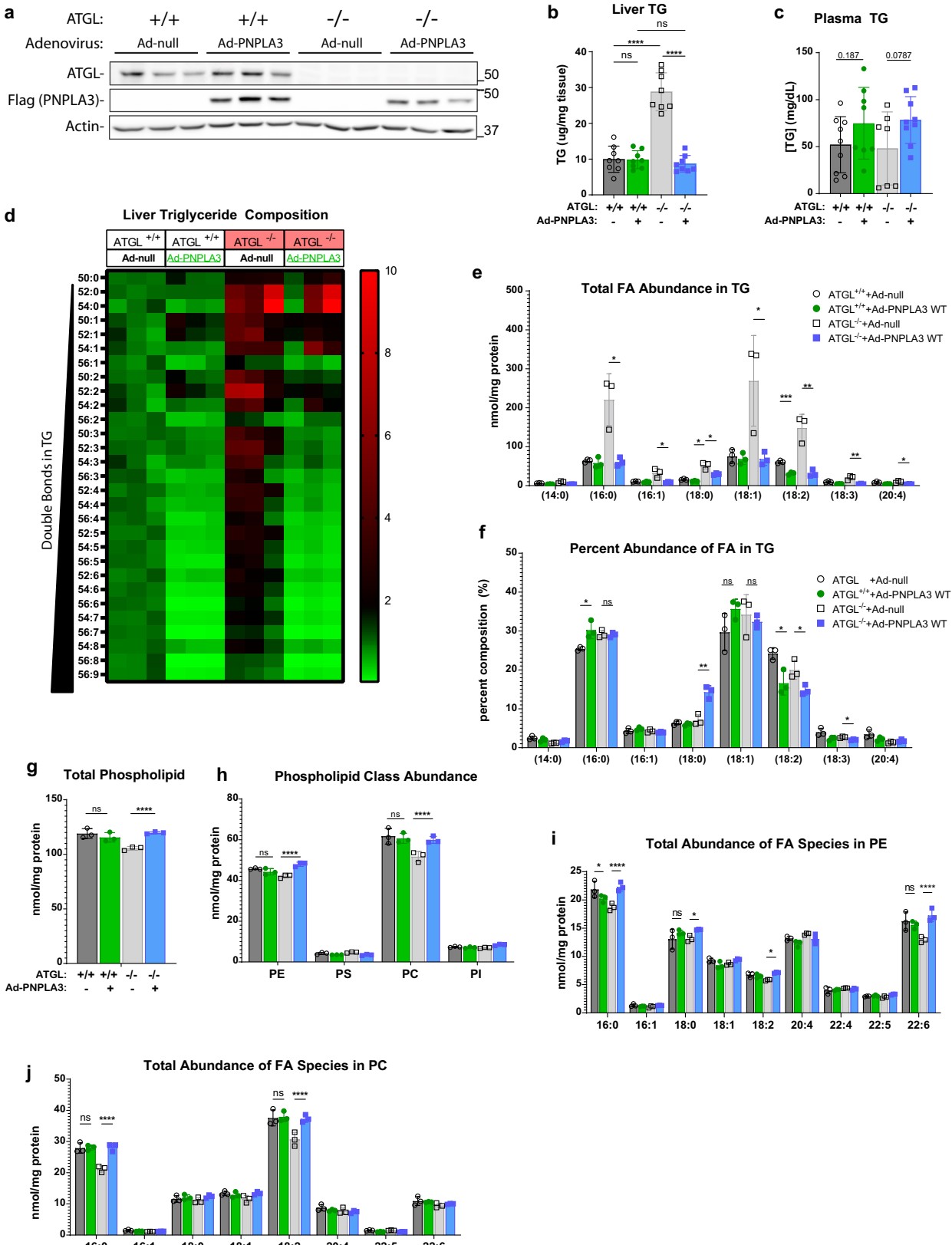

control. Notably, PNPLA3 elicited no such changes in mice on COWD without sugar in the drinking water (Supplementary Fig. 3). Together, these data suggest that PNPLA3-mediated PUFA mobilization may facilitate hepatic TG secretion under lipogenic conditions and that impaired hepatic TG secretion may be causal for aggravated steatosis when PNPLA3 is ablated.

**PNPLA3 knockdown in mice impairs mobilization of PUFAs from TG to PC during LXR agonism**

PNPLA3 expression is known to be downstream of the LXR/SREBP-1 lipogenic pathway[60]. Thus, we next sought to address whether pharmacological activation of LXR could similarly provide an appropriate setting for studying PNPLA3 function in vivo (Fig. 4a). In wild-type mice

**Fig. 2 | Adenoviral expression of PNPLA3 alleviates hepatic steatosis and reduces PUFAs in liver TG of ATGL⁻/⁻ mice.** 10-week-old WT or ATGL⁻/⁻ mice fed a chow diet were injected with Ad-null or Ad-FLAG-PNPLA3. Tissues were collected following a 12-h fast ($n = 8$/group). **a** Immunoblotting analysis of ATGL and FLAG-PNPLA3 expression in the liver. **b** Total liver TG content. **c** Total plasma TG content. **d–j** Shotgun lipidomic analysis of whole liver tissue ($n = 3$/group). **d** Heatmap indicating fold-change of individual TG species relative to control (WT mice injected with Ad-null) arranged by increasing number of double bonds. Individual FA composition of TGs presented as absolute FA abundance (**e**) and as percentage of the total FA pool (**f**). Abundance of total PLs (**g**) and major PL classes (**h**). Absolute FA abundance in PE (**i**) and PC (**j**) species. All error bars represent mean ± SD. *$p < 0.05$, **$p < 0.01$, ***$p < 0.001$ ****$p < 0.0001$, two-sided $t$ test. Specific $p$ values: **b** (ATGL + /+) Ad-null vs Ad-PNPLA3: $p = 0.938$; (Ad-null) ATGL + /+ vs ATGL −/−: $p = 9.01E-07$; (Ad-PNPLA3) ATGL + /+ vs ATGL −/−: $p = 0.356$; (ATGL−/−) Ad-null vs Ad-PNPLA3: $p = 1.09E-07$. **e** 16:0, (ATGL−/−) Ad-null vs Ad-PNPLA3: $p = 0.0149$; 16:1, (ATGL−/−) Ad-null vs Ad-PNPLA3: $p = 0.0228$; 18:0, (ATGL + /+) Ad-null vs Ad-PNPLA3: $p = 0.0176$, (ATGL−/−) Ad-null vs Ad-PNPLA3: $p = 0.0156$; 18:1, (ATGL−/−) Ad-null vs Ad-PNPLA3: $p = 0.0418$; 18:2, (ATGL + /+) Ad-null vs Ad-PNPLA3: $p = 4.33E-04$, (ATGL−/−) Ad-null vs Ad-PNPLA3: $p = 0.00483$; 18:3, (ATGL −/−) Ad-null vs Ad-PNPLA3: $p = 0.00876$; 20:4, (ATGL−/−) Ad-null vs Ad-PNPLA3: $p = 0.0117$. **f** 16:0, (ATGL + /+) Ad-null vs Ad-PNPLA3: $p = 0.0211$, (ATGL−/−) Ad-null vs Ad-PNPLA3: $p = 0.831$; 18:0, (ATGL−/−) Ad-null vs Ad-PNPLA3: $p = 0.0041$, 18:1, (ATGL + /+) Ad-null vs Ad-PNPLA3: $p = 0.126$, (ATGL−/−) Ad-null vs Ad-PNPLA3: $p = 0.581$; 18:2, (ATGL + /+) Ad-null vs Ad-PNPLA3: $p = 0.0186$, (ATGL−/−) Ad-null vs Ad-PNPLA3: $p = 0.0297$; 18:3, (ATGL−/−) Ad-null vs Ad-PNPLA3: $p = 0.0104$. **g** (ATGL + /+) Ad-null vs Ad-PNPLA3: $p = 0.383$; (ATGL−/−) Ad-null vs Ad-PNPLA3: $p = 0.000223$. **h** PE, (ATGL + /+) Ad-null vs Ad-PNPLA3: $p = 0.188$, (ATGL −/−) Ad-null vs Ad-PNPLA3: $p = 0.00170$; PC, (ATGL + /+) Ad-null vs Ad-PNPLA3: $p = 0.713$, (ATGL −/−) Ad-null vs Ad-PNPLA3: $p = 0.00934$. **i** 16:0, (ATGL + /+) Ad-null vs Ad-PNPLA3: $p = 0.158$, (ATGL−/−) Ad-null vs Ad-PNPLA3: $p = 0.00356$; 18:0, (ATGL + /+) Ad-null vs Ad-PNPLA3: $p = 0.324$, (ATGL−/−) Ad-null vs Ad-PNPLA3: $p = 0.00709$; 18:2, (ATGL−/−) Ad-null vs Ad-PNPLA3: $p = 0.000145$; 22:6(ATGL + /+) Ad-null vs Ad-PNPLA3: $p = 0.577$, (ATGL −/−) Ad-null vs Ad-PNPLA3: $p = 0.0037$. **j** 16:0, (ATGL + /+) Ad-null vs Ad-PNPLA3: $p = 0.908$, (ATGL−/−) Ad-null vs Ad-PNPLA3: $p = 0.00154$; 18:2, (ATGL + /+) Ad-null vs Ad-PNPLA3: $p = 0.841$, (ATGL−/−) Ad-null vs Ad-PNPLA3: $p = 0.0105$.

---

on COWD, injection of T0901317 (T09), a potent agonist of LXR, drastically increased both plasma and liver TG levels, consistent with previous findings[65]. Compared to those receiving control ASO, mice treated with the PNPLA3-specific ASO exhibited an almost complete loss of hepatic PNPLA3 expression (Fig. 4b). Under the COWD/T09 conditions, PNPLA3 knockdown caused a profound increase in hepatic TG accumulation and accentuated steatosis (Fig. 4c and Supplementary Fig. 4a, b) and resulted in a marked decrease in plasma TG levels (Fig. 4d). Again, loss of PNPLA3 did not appear to affect expression of lipogenic genes or ER stress markers (Supplementary Fig. 4c and Supplementary Fig. 4d). Thus, the effects of PNPLA3 ablation in response to LXR agonism phenocopy those obtained previously when lipogenesis was activated by sugar water.

To link the physiological consequences of PNPLA3 loss in vivo to its biochemical activity as a PUFA-selective TG lipase we sought to identify the compositional changes of our COWD/T09 mouse livers to identify accompanied with PNPLA3 loss. Consistent with elevated total TG levels, the absolute abundance of almost all FA species increased in liver TGs upon PNPLA3 knockdown (Fig. 4e). Compositionally, PNPLA3 knockdown caused an increased enrichment of 18:2 and a decreased enrichment of 18:1 when the amounts of FAs were expressed as a percentage of total TG-FAs (Fig. 4f). Interestingly, DAGs exhibited opposite changes in the composition of 18:1 and 18:2 (Fig. 4g). In comparison, the percentage of 16:0 remained unchanged in both TGs and DAGs. Moreover, investigating which lipid species would be most affected by PUFA retention in TGs revealed a reduction in total liver PLs (Fig. 4h). This reduction was primarily accounted for by a decrease in total PC, as total levels of PE, PI, and PS were not significantly affected (Fig. 4i). Analysis of PC acyl chain composition revealed that total PUFA, but not SFA or MUFA, content was reduced by PNPLA3 loss (Fig. 4j). In contrast, saturated or monounsaturated PC remained unchanged (Fig. 4i). Among PUFAs contained in PC, there was no difference in total amounts of 18:2, while PUFAs with longer carbon chains and more desaturation such as 20:4 and 22:6 were significantly depleted upon PNPLA3 knockdown (Fig. 4k). Taken together, these results demonstrate that loss of PNPLA3 leads to aberrant retention of PUFAs in TG and a reciprocal loss of PUFA-PC content in the liver. Our findings are consistent with PNPLA3 as a PUFA-selective TG hydrolase as opposed to acyltransferase, as free PUFAs derived from lipolysis can be sequentially elongated and further desaturated prior to incorporation into PC[66].

## VLDL-TG secretion is impaired upon PNPLA3 knockdown under lipogenic conditions

Next, we sought to directly determine the effects of PNPLA3 loss on hepatic TG secretion in mice that were fed COWD and administered T09. Following injection with Poloxamer 407 to inhibit LPL-mediated plasma TG clearance, the time-dependent TG accumulation in the plasma was significantly decreased in mice receiving the PNPLA3-specific ASO compared to those treated with the control ASO (Fig. 5a, b). We then subjected these plasma samples to lipidomic analysis to assess whether the FA enrichment differences observed in the liver were reflected in the secreted lipid pool. Absolute levels of all fatty acid species were depleted in secreted TG and PC upon PNPLA3 knockdown (Fig. 5c, e), consistent with disrupted hepatic TG secretion, however the PUFA-TG enrichment observed in the liver was not reflected in the secreted lipids (Fig. 5d, f). Notably PC containing 20:4 and 22:6, the same FA species specifically depleted in PNPLA3 KD livers, were also specifically depleted from plasma PC, though these differences did not eclipse statistical significance ($p = 0.066$, $p = 0.090$ respectively).

Given the importance of PL desaturation in regulating TG packaging and the size of VLDLs, we asked whether impaired secretion of large-sized VLDLs could be the mechanism by which loss of PNPLA3 impairs hepatic TG output, thus promoting intrahepatic TG retention and steatosis. As revealed by fast protein liquid chromatography (FPLC) analysis of plasma lipoprotein particles, knockdown of PNPLA3 caused a profound reduction in the TG content of the VLDL fraction (Fig. 5g) along with decreased cholesterol content in the HDL fraction (Fig. 5h). Elevated TG levels in the VLDL fraction appeared to reflect the size of VLDL particles as plasma levels of apoB-48 and apoB-100 were both unaffected by PNPLA3 knockdown (Fig. 5i). Subsequently, direct examination using transmission electron microscopy (TEM) confirmed the size reduction of VLDL particles obtained from the PNPLA3 knockdown mice (Fig. 5j, k). Altogether, these data demonstrate that PNPLA3 is critically involved in regulating the secretion of large-sized VLDL particles during lipogenic stimulation.

## PNPLA3 facilitates flux of PUFAs from intracellular TG to secreted PL

We next determined how loss of PNPLA3 would impact hepatic FA flux and partitioning. To verify the cell-autonomous effects of PNPLA3, we knocked down PNPLA3 expression in isolated primary mouse hepatocytes with ASO (Fig. 6a). After pretreatment with FAs in low glucose media to promote TG-LD accumulation, cells were switched to FA-free, high glucose media to simulate lipogenic conditions induced by high sucrose refeeding following fasting. Compared to control ASO-transfected cells, PNPLA3 ASO-transfected cells exhibited no differences in LD accumulation upon pretreatment with either OA or LA in low glucose conditions (Supplementary Fig. 5). In control cells, replacement in FA-free, high glucose medium

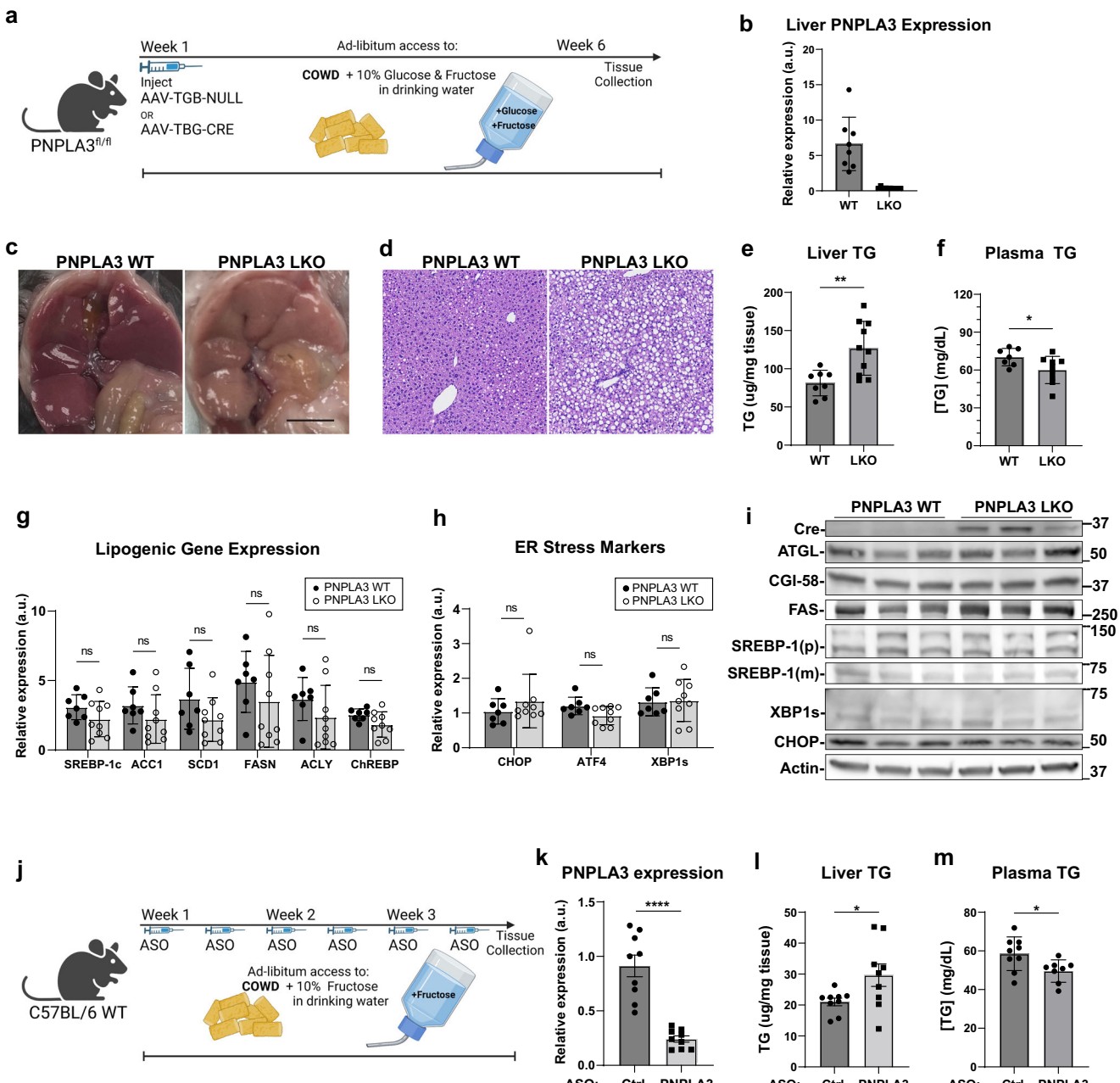

**Fig. 3 | PNPLA3 loss induces hepatic steatosis but ameliorates hypertriglyceridemia in mice challenged with a PUFA-enriched diet and sugar water.**
**a** Experimental scheme (Biorender) for creation and treatment of PNPLA3-LKO mice. 8–10-week-old PNPLA3[fl/fl] mice injected with either AAV-TBG-Null (*n* = 8) or AAV-TBG-Cre (*n* = 10) were fed COWD and given access to fructose/glucose in drinking water for 6 weeks. **b** Liver PNPLA3 mRNA expression as analyzed by qPCR. **c** Gross appearance of liver (scale bar = 1 cm). **d** Hematoxylin and eosin (H&E) staining of liver sections. **e** Total liver TG content. **f** Total plasma TG content. **g** Liver mRNA expression of lipogenic genes by qPCR. **h** Liver mRNA expression of ER stress response genes by qPCR. **i** Immunoblotting analysis of indicated lipolysis, lipogenesis, and ER-stress associated proteins in liver. **j** Experimental scheme (Biorender) for PNPLA3 ASO experiment. 10-week-old WT mice fed COWD and given fructose drinking water were treated with control or PNPLA3 ASO biweekly for 3 weeks (*n* = 9/group). **k** Liver PNPLA3 mRNA expression in ASO mice by qPCR. **l** Total liver TG content. **m** Total plasma TG content in ASO mice. Data presented in (**e**–**h**) and (**k**–**m**) are independent replicates derived from individual mice. Data are represented as mean ± SD. **p* < 0.05, ***p* < 0.01, two-sided t test. Specific *p* values: **e** *p* = 0.00410. **f** *p* = 0.0443. **k** *p* = 7.73E-06. **l** *p* = 0.0389. **m** *p* = 0.0272.

induced a more drastic degradation of LDs formed by LA than OA (Fig. 6b, c). However, the degradation of LA-formed LDs was largely prevented when PNPLA3 was knocked down, indicating a specific role of endogenous PNPLA3 in hydrolyzing PUFA-TGs under lipogenic conditions. To a lesser degree, PNPLA3 knockdown also prevented LD degradation in cells treated with OA (Fig. 6b, c), likely due to disrupted secretion.

Transfer of PUFAs into PLs is critical for synthesis and secretion of large-sized TG-rich VLDLs[21,67]. Since de novo FA synthesis does not produce PUFAs, we hypothesized that PUFAs derived from PNPLA3-mediated lipolysis would contribute directly to PUFA-containing phospholipid synthesis in support of hepatic TG secretion. To test the effect of PNPLA3 knockdown on differential FA partitioning and flux to intracellular and secreted lipids, we pulse labeled primary hepatocytes with either [3]H-OA or [14]C-LA in low-glucose medium that contained equal amounts of unlabeled OA and LA. Following a chase period in high-glucose, FA-free medium, we extracted and analyzed cellular and secreted lipids as well as the incorporation of radiolabeled

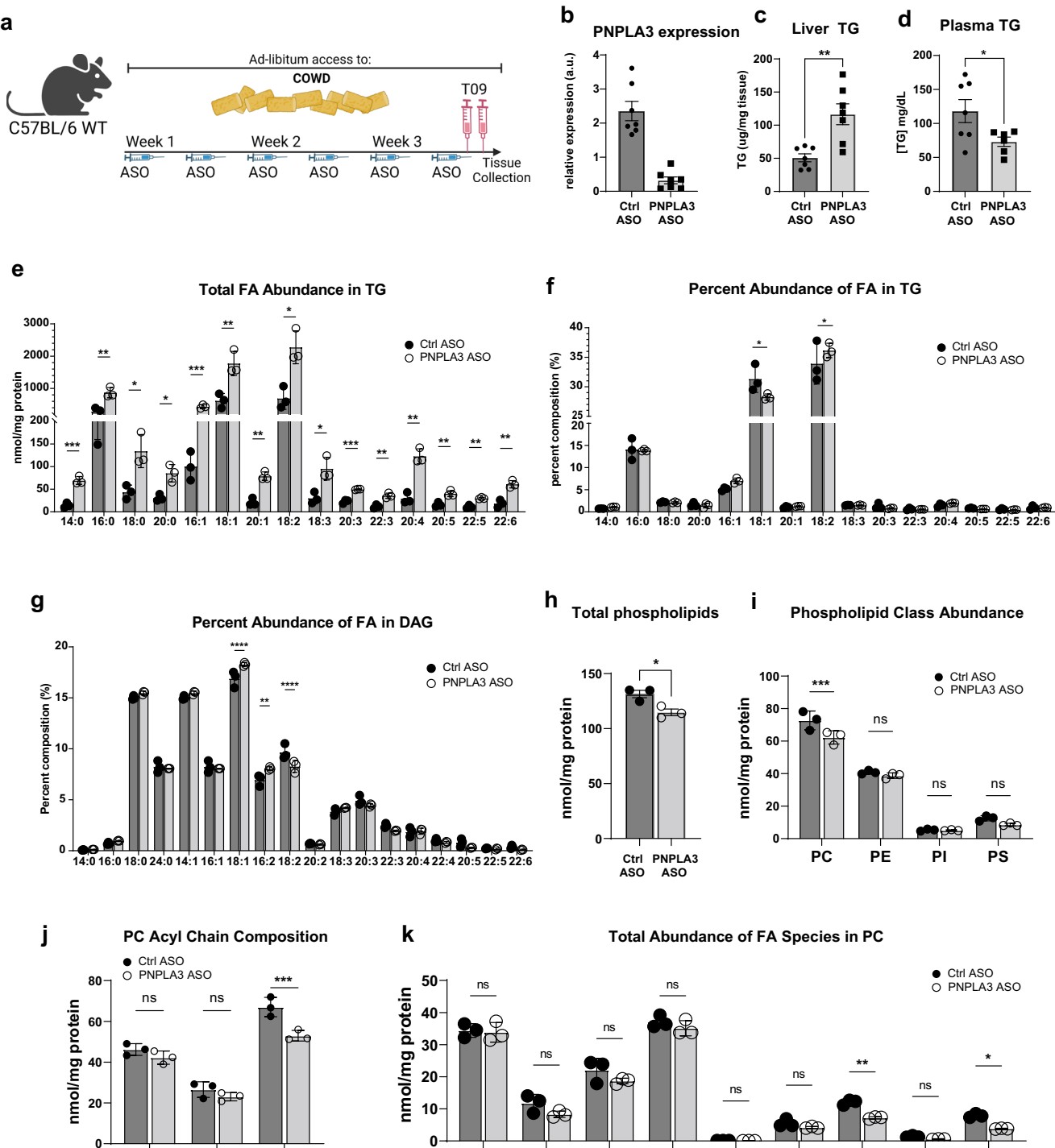

**Fig. 4 | PNPLA3 depletion impairs mobilization of PUFA from TG to PC upon LXR agonism.** 10-week-old WT mice fed COWD were treated with control or PNPLA3 ASO for 3 weeks. Both groups were treated with the LXR agonist T09 prior to tissue collection (*n* = 7/group). **a** Experimental scheme (Biorender). **b** Liver PNPLA3 mRNA expression. **c** Total liver TG content. **d** Total plasma TG content. **e**–**k** Shotgun lipidomics was performed on whole liver tissue (*n* = 3/group). Individual FA species composition of TG presented as absolute FA abundance (**e**) and as percentage of the total FA pool (**f**). **g** Percentage of total FA pool in DAG. Abundance of total PLs (**h**) and major PL classes (**i**). **j** Total abundance of SFA, MFA, and PUFA acyl chains in PC. **k** Absolute FA abundance in PC. Data presented as

mean ± SD *$p < 0.05$, **$p < 0.01$, ***$p < 0.001$ ****$p < 0.0001$, two-sided *t* test. Specific *p* values: **c** $p = 0.00211$. **d** $p = 0.0402$. **e** 14:0, $p = 7.24E\text{-}04$; 16:0, $p = 0.0160$; 18:0, $p = 0.00681$; 20:0, $p = 0.00296$; 16:1, $p = 0.00742$; 18:1, $p = 0.0166$; 20:1, $p = 0.0101$; 18:2, $p = 0.00998$; 18:3, $p = 0.0110$; 20:3, $p = 1.06E\text{-}04$; 22:3, $p = 8.61E\text{-}04$; 20:4, $p = 4.13E\text{-}04$; 20:5, $p = 0.00136$; 22:5, $p = 0.00113$; 22:6, $p = 0.00114$. **f** 18:1, $p = 0.509$; 18:2, $p = 0.0460$. **g** 18:1, $p = 0.00226$; 16:2, $p = 0.0310$; 18:2, $p = 0.0418$. **h** $p = 0.0214$. **i** PC, $p = 0.0459$. **j** SFA, $p = 0.186$; MUFA, $p = 0.236$; PUFA, $p = 0.0108$. **k** 16:0, $p = 0.805$; 18:0, $p = 0.114$; 18:1, $p = 0.192$; 18:2, $p = 0.327$; 18:3, $p = 0.00367$; 20:3, $p = 0.140$; 20:4, $p = 6.46E\text{-}04$; 22:5, $p = 0.0104$; 22:6, $p = 2.96E\text{-}04$.

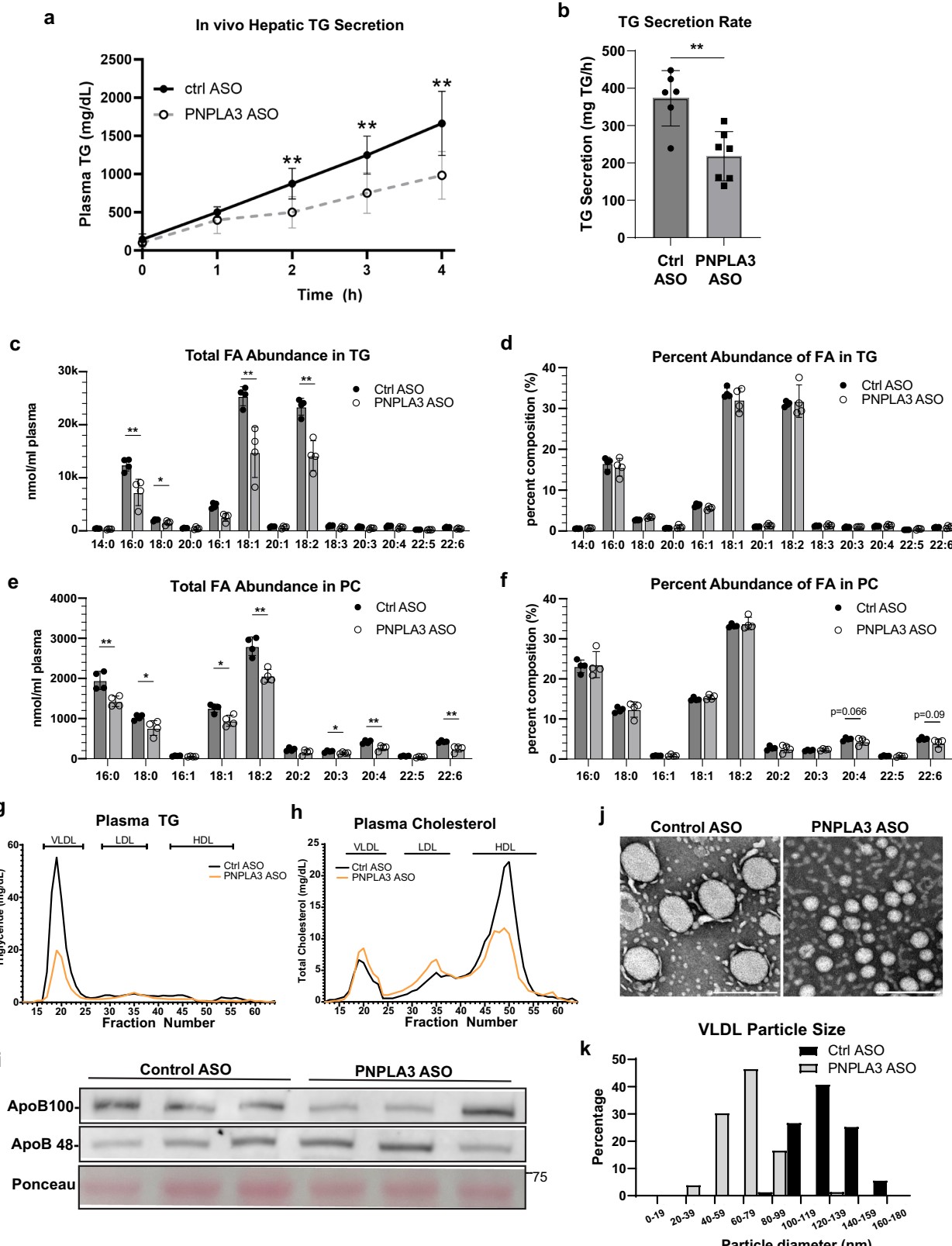

tracers into TGs and PLs of these two lipid pools. As shown in Fig. 6d, e, cells with PNPLA3 knockdown exhibited an increased intracellular TG retention along with a decreased TG secretion.

While it reduced relative incorporation of both OA and LA tracers into cellular PLs (Fig. 6f), PNPLA3 knockdown only led to decreased incorporation of LA tracer into secreted PLs (Fig. 6g). In addition, when

cells were labeled exclusively with either OA ($^3$H-OA+unlabeled OA) or LA ($^{14}$C-LA+unlabeled LA), knockdown of PNPLA3 resulted in decreased incorporation of LA to PLs relative to TGs in secreted lipids, while OA-PL incorporation was trending higher ($p = 0.2332$) (Fig. 6h, i). Collectively, these results suggest that PNPLA3 regulates TG secretion via mobilizing PUFAs from intracellular TG to both intracellular PLs

**Fig. 5 | PNPLA3 depletion reduces VLDL particle size and TG content as well as impairs hepatic TG secretion upon LXR agonism. a** Representative time course of hepatic TG secretion in mice upon administration of Poloxomer-407. Priorly, 10-week-old WT mice were fed COWD and treated with control (*n* = 6) or PNPLA3 ASO (*n* = 7) and administered T09 for 3 weeks. **b** Data from (**a**) represented as secretion rate per hour. **c–f** Shotgun lipidomics was performed on plasma samples from the 3 h timepoint in (**a**) (*n* = 4/group). Individual FA species composition of TG presented as absolute FA abundance (**c**) and as percentage of the total FA pool (**d**). Individual FA species composition of PC presented as absolute FA abundance (**e**) and as percentage of the total FA pool (**f**). Plasma samples pooled from control or

PNPLA3 ASO mice administered T09 were fractionated by FPLC (4-5 mice/group). Lipoprotein fractions were measured for the content of TG (**g**) and cholesterol (**h**). **i** Plasma samples were immunoblotted for ApoB-100 and ApoB-48. **j** EM analysis of lipoprotein particles from control and PNPLA3 ASO mice +T09. **k** Quantification of VLDL particle size distribution of (**J**). Scale bar = 200 nm. Data presented as mean ± SD *$p$ < 0.05, **$p$ < 0.01, two-sided *t* test. Specific p values: (**a**) 1 h, $p$ = 0.214; 2 h, $p$ = 0.00661; 3 h, $p$ = 0.00544; 4 h, $p$ = 0.00643. (**b**) $p$ = 0.00212. (**c**) 16:0, $p$ = 0.00923; 18:0, $p$ = 0.0452; 16:1, $p$ = 0.00309; 18:1, $p$ = 0.00587; 18:2, $p$ = 0.00125. **e** 16:0, $p$ = 0.00718; 18:0, $p$ = 0.0344; 18:1, $p$ = 0.0109; 18:2, $p$ = 0.00194; 20:3, $p$ = 0.0456; 20:4, $p$ = 0.00966; 22:6, $p$ = 0.00454.

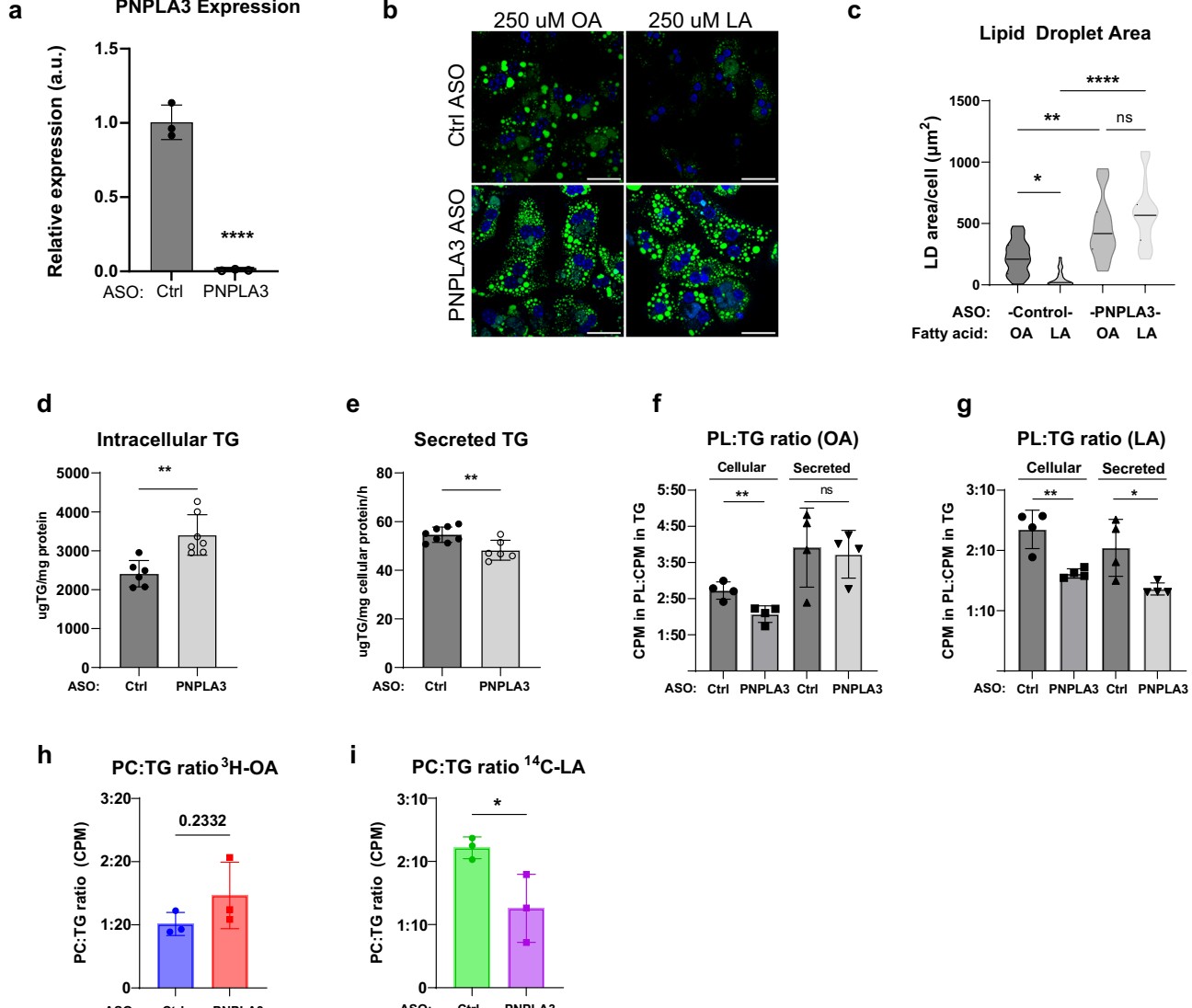

**Fig. 6 | PNPLA3 mediates PUFA mobilization from intracellular TG to secreted PL. a** qPCR of PNPLA3 expression in primary hepatocytes transfected with control or PNPLA3 ASO (*n* = 3/group). **b** BODIPY staining of LDs in cells after 4 h of lipid deprivation in high glucose medium following treatment with 250 μM OA or LA in low glucose medium. **c** Quantification of LD area in (**b**). Primary hepatocytes were pretreated with a mixture of unlabeled 125 μM OA + 125 μM LA (**n** = 8/ASO group). Following 24 h in FA-free media, lipids were extracted from cells (**d**) and conditioned media (**e**) for quantitative TG analysis. Primary hepatocytes were pretreated with a mixture of unlabeled 125 μM OA + 125 μM LA containing either [3]H-OA or [14]C-LA. After a 24-h chase period, cellular and secreted lipids were

analyzed for the ratio of [3]H-OA (**f**) or [14]C-LA (**g**) in PL vs TG (*n* = 4/group). Primary hepatocytes were pretreated with either 125 μM of unlabeled OA containing [3]H-OA (**h**) or 125 μM of unlabeled LA containing [14]C-LA (**i**). After a 24 h chase period, secreted lipids were analyzed for ratio of radioactivity in PL vs.TG (*n* = 3/group). Scale bar = 50 μm. Box-plot elements: center line, median; box limits, upper and lower quartiles. Data presented as mean ± SD *$p$ < 0.05, **$p$ < 0.01, ****$p$ < 0.0001, two-sided *t* test. Specific p values: **c** (Ctrl ASO) OA vs LA, $p$ = 2.23E-06; (PNPLA3 ASO) OA vs LA, $p$ = 0.177; (OA) Ctrl ASO vs PNPLA3 ASO, $p$ = 0.00239; (LA) Ctrl ASO vs PNPLA3 ASO, $p$ < 1E-10. **d** $p$ = 0.00213. **e** $p$ = 0.00666. **f** Cellular, $p$ = 0.00775; Secreted, $p$ = 0.783. **g** Cellular, $p$ = 0.00437; Secreted, $p$ = 0.0300. **i** $p$ = 0.0424.

and PLs of secreted lipoproteins, the disruption of which significantly impairs hepatic TG secretion.

### The PNPLA3 I148M mutant impairs PUFA mobilization and inhibits TG secretion

Having established a significant physiological role for wild-type PNPLA3, we next sought to evaluate the impact of the I148M mutant in hepatic TG secretion. We first expressed either wild-type PNPLA3 or I148M mutant in primary mouse hepatocytes using recombinant adenoviruses to evaluate the impact on TG secretion in vitro. To this end, cells were treated with a mixture of OA and LA along with $^3$H-glycerol to allow measurement of TGs secreted into the conditioned media. Whereas the expression of the wild-type protein trended toward increasing TG secretion relative to controls ($p = 0.12$), primary hepatocytes expressing the I148M mutant showed impaired TG secretion in comparison to those expressing the WT protein (Fig. 7a). In vivo, adenoviral delivery of PNPLA3 I148M induced significant hepatic TG accumulation (Fig. 7b), consistent with prior findings[49]. Consistently, expression of PNPLA3 I148M increased the absolute abundance of most FA species in the liver (Fig. 7c). It did not, however, change the 18:2 FA fraction of the TG (Fig. 7d), in contrast to expression of the wild-type PNPLA3 (Fig. 2f). Interestingly, 20:4 was significantly depleted from liver TGs upon I148M expression both in absolute and fractional amounts (Fig. 7c, d). Furthermore, total phospholipids (Fig. 7e), PC and PE in particular (Fig. 7f), were reduced upon expression of PNPLA3 I148M. Compositionally, 18:2 and 20:4 and were specifically depleted from PC and PE (Fig. 7g–j), suggesting that expression of the I148M mutant impairs PUFA mobilization from TG to PLs.

It has been previously shown that KI mice harboring the I148M mutation at the endogenous PNPLA3 locus develop steatosis upon a high-sucrose diet challenge[61,68]. This is in contrast to PNPLA3 I148M overexpression which presents hepatic steatosis absent any dietary challenge. Therefore, we sought to utilize PNPLA3 KI mice to study the effect of endogenously expressed PNPLA3 I148M on hepatic TG secretion. In vitro, primary hepatocytes isolated from PNPLA3 KI mice had significantly less TG secretion when compared to WT hepatocytes (Fig. 7k). Liver and plasma TG parameters in PNPLA3 KI mice fed COWD and given access to fructose and glucose in drinking water for 8 weeks recapitulated the impact of liver specific PNPLA3 loss in vivo (Fig. 7l, m). To assess the effect of PNPLA3 I148M on TG secretion directly, we fed mice COWD for 3 weeks with access to fructose and glucose in the drinking water, and then assayed hepatic TG secretion following injection with Poloxamer-407. Importantly, PNPLA3 KI mice exhibited a significantly reduced rate of TG secretion compared to the wild-type littermates (Fig. 7n, o), consistent with the effects observed in PNPLA3 ablation models. Taken together, these data suggest the PNPLA3 I148M mutation is a loss-of-function for PUFA-TG lipase activity, resulting in impaired hepatic TG secretion during lipogenic stimulation.

### Discussion

Despite the significant association of its I148M variant with MAFLD, the physiological relevance of wild-type PNPLA3 in the regulation of hepatic lipid homeostasis remains elusive. During lipogenic stimulation, de novo synthesized FAs are directed toward *TG synthesis* for incorporation into VLDL. In this context, the current study demonstrates that PNPLA3, whose expression increases in response to carbohydrate feeding, is a TG hydrolase with substrate specificity toward PUFA-containing TGs and thereby plays a key role in balancing hepatic TG storage and secretion (Supplementary Fig. 6). Our experiments using biochemical techniques, cultured cells, and mice show that PNPLA3 specifically degrades PUFA-induced LDs in a CGI-58-dependent but ATGL-independent manner. The facts that PNPLA3 overexpression in ATGL$^{-/-}$ mice increased PUFA enrichment in the liver PLs as well as total plasma TG levels suggest that PNPLA3 functions to mediate PUFA flux to PLs in support of hepatic TG secretion. This

conclusion is further supported by the findings that both loss-of-function animal models, namely PNPLA3-LKO mice and PNPLA3 knockdown mice, displayed decreased plasma TG levels and aggravated liver steatosis upon lipogenic stimulation following treatment with the PUFA-enriched COWD. Furthermore, pulse-chase experiment using primary hepatocytes provides evidence that PNPLA3 promotes channeling of PUFAs from intracellular TG to PLs. To the best of our knowledge, the current study represents the first ever effort to establish the specific metabolic conditions under which wild-type PNPLA3 exerts a significant physiological function. Importantly, the I148M-KI mice largely phenocopied the knockout mice and exhibited impaired hepatic TG secretion under the same dietary conditions. This loss of function under the condition of lipogenic stimulation provides a potential explanation for the increased susceptibility of patients carrying the I148M variant to the development of liver steatosis and MAFLD.

The biochemical function of PNPLA3 has been the subject of investigation over the past decade, but its TG hydrolase activity was not consistently established toward either triolein substate in vitro or OA-induced LDs in cells[47,55,69]. As PNPLA3 was shown to be capable of inhibiting ATGL by competing for CGI-58[54], we thought that previous efforts could be confounded by the presence of ATGL, the predominant TG hydrolase especially in lipolytic conditions. By utilizing ATGL$^{-/-}$ HeLa cells and LDs isolated from primary hepatocytes, we obtained data that decisively demonstrate the ATGL-independent ability of PNPLA3 to selectively degrade PUFA-LDs. Four pieces of evidence are in support of the TG hydrolase activity of PNPLA3 being responsible for this LD-degrading capability. Firstly, the action of PNPLA3 is dependent on CGI-58 as demonstrated by the reduced LD clearance in CGI-58-KO cells. Secondly, alanine substitution of the active serine (Ser47) in the hydrolytic GXSXG motif of PNPLA3 completely abolished the LD degradation upon PNPLA3 expression. Thirdly, coexpression of G0S2, a known inhibitors of the TG hydrolase activity of ATGL[45], was sufficient to block PNPLA3-mediated LD degradation. Finally, PNPLA3 presence on LDs isolated from primary hepatocytes led to significant FA release, particularly when LDs were enriched in LA. Curiously, a recent study showed that PNPLA3 deletion in human epidermoid carcinoma-derived A431 cells resulted in accumulation of LA-induced LDs without affecting the rate of TG hydrolysis[55]. It was speculated that PNPLA3 might act as a DAG transacylase or DAG hydrolase instead of a TG hydrolase. While we cannot exclude the possibility of PNPLA3 directly acting on DAG substrates, our further findings that PNPLA3 deletion caused opposite compositional changes in hepatic TG and DAG species strongly support its role as a catabolic lipase for PUFA-containing TGs. In addition to demonstrating direct FA release from isolated LDs, the fact that PNPLA3 knockdown caused decreased abundance of very long-chain FAs such as 20:4 and 22:6 in PC without changes in 18:2, suggests that free PUFAs derived from PNPLA3-mediated lipolysis may be elongated and desaturated before their incorporation into PC rather than directly transferred to lysophospholipids.

Our results establish a critical role of PNPLA3 in the broader context of lipogenesis. Specifically, we show that loss of PNPLA3 during lipogenic stimulation impaired PUFA flux to PLs, resulting in diminished VLDL-TG secretion and increased hepatic TG accumulation and steatosis. Previously, *PNPLA3* knockout failed to produce detectable metabolic phenotypes in mice maintained on chow, high-fat, or high-sucrose diets[50,51], prompting the conclusion that loss of PNPLA3 function does not contribute to the pathogenesis of fatty liver disease, at least in mice. We think that one possible explanation as to why hepatic steatosis upon PNPLA3 loss was observed in the current study is in the specific lipogenic conditions. The substitution of corn oil for butter in the Western diet should sufficiently enrich the hepatic PUFA content in TGs during the feeding stage when lipogenic activity is low. Upon removal of the solid fat diet, continuous access to the glucose/fructose

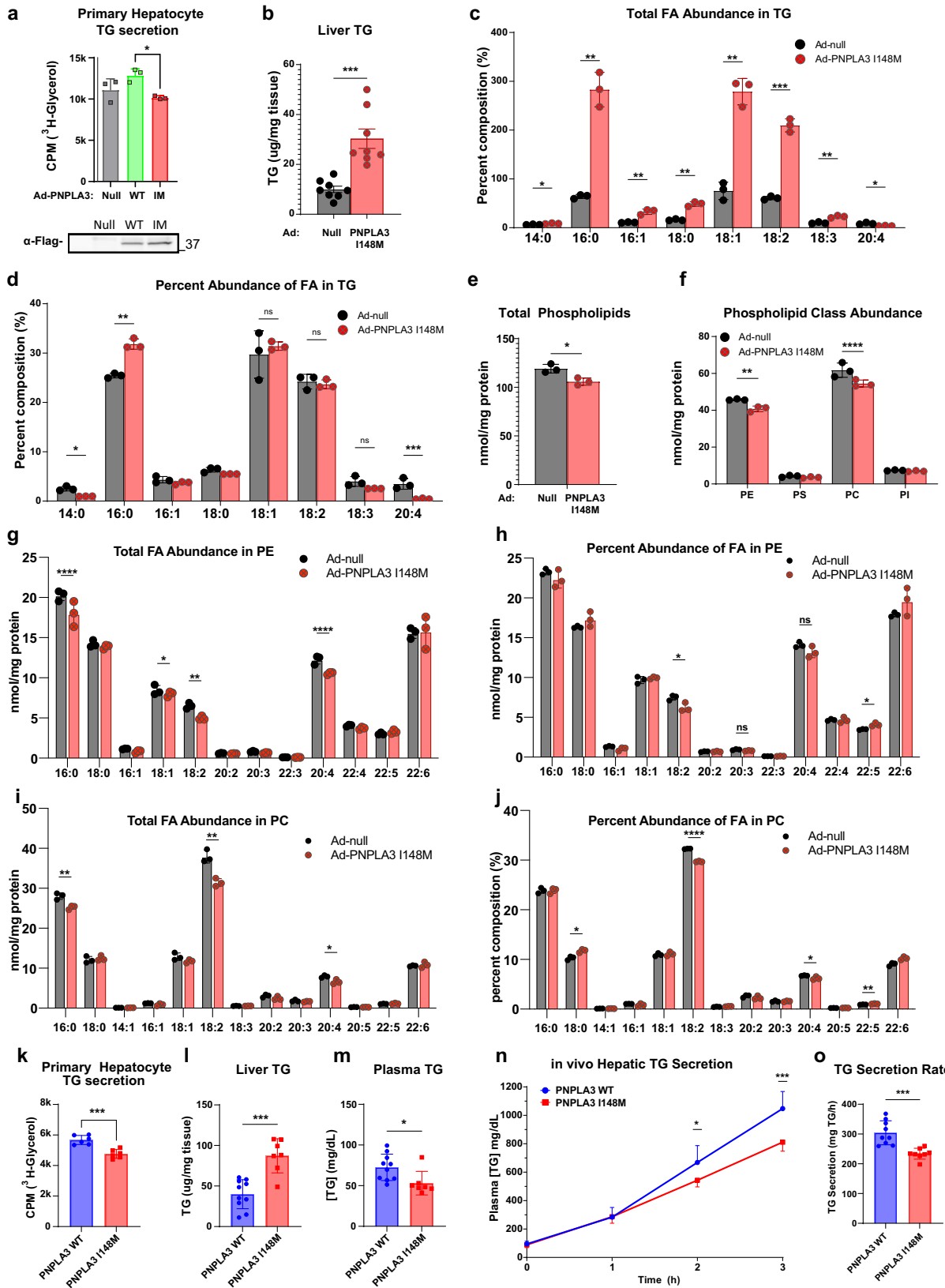

drinking water would then lead to the activation of hepatic lipogenesis, which becomes the main driver of VLDL synthesis and secretion. The dependence of this model for PNPLA3 function on lipogenic stimulation is underscored by the absence of any plasma or liver TG differences with PNPLA3 knockdown when we did not provide a lipogenic stimulus (Supplementary Fig. 3), consistent with prior reports[50].

A retrospective examination of the lipidomic profile of PNPLA3 LKO mice given access to glucose/fructose in their drinking water revealed no substantial changes in PUFA enrichment of either TG or phospholipids. It is possible that acute (LXR agonism) and chronic (sugar-infused drinking water) lipogenic stimulations might differentially influence the mechanism of PNPLA3-mediated hepatic TG

**Fig. 7 | Overexpression or knockin of PNPLA3-I148M in mice impairs hepatic TG secretion. a** TG secretion from primary hepatocytes isolated from WT mice, infected with control, PNPLA3 WT or PNPLA3 I148M expressing adenovirus, and treated with 250 µM 1:1 OA:LA and 25 µM glycerol containing ³H-glycerol (*n* = 3/ group). **b–h** 10-week-old WT mice fed a chow diet were injected with either Ad-null or Ad-PNPLA3-I148M. Tissues were collected following a 12-h fast for lipid (*n* = 8/ group) and lipidomic analysis (*n* = 3/group). **b** Total liver TG content. Individual FA species composition of TG presented as absolute FA abundance (**c**) and as percentage of the total FA pool (**d**). **e** Abundance of total PLs. **f** Abundance of major PL classes. Individual FA species composition of PE presented as absolute FA abundance (**g**) and as percentage of the total FA pool (**h**). Individual FA species composition of PC presented as absolute FA abundance (**i**) and as percentage of the total FA pool (**j**). **k** TG secretion from primary hepatocytes isolated from WT or PNPLA3-I148M-KI mice following treatment with 250 µM 1:1 OA:LA and 25 µM glycerol containing ³H-glycerol (*n* = 6/group). **l–o** 9-12-wk-old WT (*n* = 10) and I148M-

KI (*n* = 7) mice were fed COWD and given access to fructose/glucose in drinking water for 3 weeks. **l** Total liver TG content. **m** Total plasma TG content. **n** Representative time course of hepatic TG secretion in WT (*n* = 10) and PNPLA3-I148M-KI (*n* = 7) mice upon administration of Poloxomer-407 (**o**) Data from (**n**) represented as secretion rate per hour. All error bars represent mean ± SD. *\*p < 0.05, \*\*p < 0.01, \*\*\*p < 0.001 \*\*\*\*p < 0.0001*, two-sided *t* test. Specific *p* values: **a** *p* = 0.0445. **b** *p* = 2.16E-04. **c** 14:0, *p* = 8.65E-04; 16:0, *p* = 5.10E-04; 16:1, *p* = .00215; 18:0, *p* = 1.75E-04; 18:1, *p* = 2.91E-04; 18:2, *p* = 2.36E-05; 18:3, *p* = 1.89E-04; 20:4, *p* = 0.434. **d** 14:0, *p* = 0.00386; 16:0, *p* = 0.6.23E-04; 18:1, *p* = 0.573; 18:2, *p* = 0.611; 18:3, *p* = 0.0700; 20:4, *p* = 0.00793. **e** *p* = 0.0406. **f** PE, *p* = 0.0487; PC, *p* = 0.0258. **g** 16:0, *p* = 0.0801; 18:2, *p* = 0.00376; 20:4, *p* = 0.00670. **h** 18:2, *p* = 0.0277; 20:3, *p* = 0.0618; 20:4, *p* = 0.114; 22:5, *p* = 0.0110. **i** 16:0, *p* = 0.00707; 18:1, *p* = 0.126; 18:2, *p* = 0.00415; 20:4, *p* = 0.021. **j** 18:0, *p* = 0.00761; 18:2, *p* = 3.43E-06; 20:4, *p* = 0.0429; 22:5, *p* = 0.00981. **k** *p* = 2.96E-04. **l** *p* = 1.60E-04. **m** *p* = 0.0223. **n** 1 h, *p* = 0.731; 2 h, *p* = 0.0269; 3 h, *p* = 4.19E-04. **o** *p* = 0.000522.

secretion. Under chronic stimulation, the negative impact of PNPLA3 loss on VLDL may be relatively more subtle, yet it could lead to hepatic steatosis over time. Additionally, in PNPLA3 LKO mice, adipose-derived PUFAs may contribute to hepatic phospholipid desaturation after a 12-hour solid food fast, resulting in less pronounced differences in hepatic FA compositions at both bulk tissue and plasma levels. Conversely, our knockdown approach involved unconjugated antisense oligonucleotide (ASO), which may effectively target PNPLA3 in the liver as well as in adipose tissue. Nevertheless, hepatic steatosis caused by the loss of PNPLA3 is likely the outcome of impaired VLDL lipidation and thus TG secretion.

While de novo FAs can be used for the synthesis of saturated and monounsaturated lipids of VLDL, our data from the pulse-chase experiment strongly suggest that PNPLA3 is instrumental in mobilizing PUFA from the pre-existing cytosolic TG pool for the synthesis of polyunsaturated PLs to support VLDL lipidation. Data from our pulse-chase experiment demonstrated that PNPLA3 loss significantly impairs PL generation, especially of PUFA-PLs, and reduces hepatic TG secretion. In vivo, our lipidomic data indicate a significant role of PNPLA3 in the intrahepatic mobilization of PUFAs from TGs to phosphatidylcholine. This effect is accompanied by a reduced total incorporation of PUFAs into both VLDL-TGs and VLDL-phospholipids. The absence of significant compositional changes in VLDL lipids suggests that PNPLA3-mediated desaturation of endoplasmic reticulum (ER) or Golgi membranes facilitates overall VLDL lipidation rather than the incorporation of specific PL species onto VLDL. The absence of a known mechanism for selectively transferring PUFA-phospholipids onto VLDL along with the established impact of membrane PUFAs on MTTP activity[22] further support this interpretation. Interestingly, poorly lipidated VLDL lacking differences in PUFA-PC composition has also been observed in human carriers of the steatosis-associated TM6SF2 E167K variant[24,70].

Previous studies have demonstrated that LPCAT3 selectively facilitates PUFA incorporation into PCs, which is necessary for secretion of large-sized VLDL and maintaining hepatic TGs at low levels[21,22]. Although ablation of PNPLA3 appears to produce similar metabolic effects to those observed with LPCAT3 knockout, the underlying mechanisms may differ slightly. While the function of LPCAT3-mediated membrane desaturation enables VLDL-TG secretion, perhaps in coordination with PNPLA3, LPCAT3 was suggested to ease ER stress and facilitate post-translational activation of SREBP-1c[64,71], no alterations in the expression of ER stress markers or SREBP-1c and its target genes were observed with PNPLA3 disruption. These findings rule out the possibility of excess synthesis causing hepatic steatosis in *PNPLA3*-LKO or knockdown mice. Instead, PNPLA3 ablation appears to only impact VLDL lipidation and thus size gain but not secretion, as the reduction in the plasma VLDL-TG content was not accompanied by any decrease in ApoB secretion. Of note, like PNPLA3-I148M, the E167K variant of TM6SF2 is also associated with hepatic steatosis and

reduced levels of plasma TG in humans. Loss of *TM6SF2* in mice similarly results in decreased VLDL lipidation without alteration of ApoB synthesis or secretion[72–74]. It was reported that incorporation of PUFAs into TGs and PCs was decreased in *TM6SF2* knockdown hepatoma cells[24,67]. However, only marginal differences in the PUFA content of hepatic and plasma lipids were observed in wild-type and *TM6SF2* knockout mice[72]. Thus, it remains to be determined whether PNPLA3 and TM6SF2 are functionally linked in the control of VLDL lipidation and secretion.

Under the conditions of COWD and fructose/glucose drink water, I148M-KI mice showed decreased plasma TG levels along with diminished hepatic TG secretion, suggesting a loss of function by the I148M variant. However, our data do not necessarily conflict with the ATGL-dependent, gain of function paradigm proposed recently for the I148M variant[54]. While wild-type PNPLA3 is normally expressed in the fed state, the I148M mutant protein is known to be more stable and thus could act under fasting conditions as an ATGL inhibitor via binding to CGI-58. Thus, we hypothesize that loss of function to facilitate TG secretion under lipogenic conditions and gain of function as an ATGL inhibitor under lipolytic conditions both contribute to the effects of the I148M variant on the development of hepatic steatosis and MAFLD. To this end, the contributions of a loss-of-function paradigm to the development of MAFLD are consistent with two clinical observations that seem to be unaccounted for on a strictly gain-of-function disease model. First, numerous clinical investigations have demonstrated the increased association between PNPLA3 and MAFLD in human cohorts consuming high amounts of refined sugar[75–77], including the consumption of sugar sweetened beverages. Second, recent evidence suggests the I148M variant may provide an antihyperlipidemic effect in individuals with insulin resistance or obesity[57,78]. Our findings that phenotypes associated with PNPLA3 loss require a strong lipogenic stimulus (Fig. 4a–d and Supplementary Fig. 3) and that PNPLA3 facilitates VLDL-TG secretion (Fig. 5c) seem to account for these clinical observations.

In conclusion, our findings demonstrate that PNPLA3 selectively degrades PUFA-TGs in an ATGL-independent manner. In the context of a broad lipogenic response, the enzyme facilitates PUFA integration into PLs in support of VLDL-TG secretion. The I148M variant of PNPLA3 is less active than the wild-type enzyme, suggesting that hepatic steatosis results, at least in part, from a loss of function and decreased hepatic TG secretion. Finally, our results implicate the possibility that I148M variant may exert a cardioprotective impact on carriers under specific nutritional conditions such as those including high content of polyunsaturated fat in combination with excess consumption of sugary beverages.

## Methods
### Reagents
The following antibodies were used: Rabbit anti-c-Myc (Cell Signaling, #2278, 1:1000 dilution); Mouse anti-FLAG M2 (Sigma, #F1804,

1:1000 dilution); Mouse anti-Actin (Sigma, #A1978, clone AC-15, 1:10,000 dilution); Rabbit anti-ATGL (Cell Signaling, #2138, 1:500 dilution); Rabbit anti-Cre (Cell Signaling, #15036, 1:2000 dilution); Rabbit anti-ABHD5 (proteintech, #12201, 1:1000 dilution); Rabbit anti-FAS (Cell Signaling, #3180, 1:1000 dilution); Mouse anti-SREBP-1 (Santa Cruz Biotechnology, sc-13551, 1:500 dilution); Rabbit anti-XBP-1s (Cell Signaling, #40435 1:1000 dilution); Rabbit anti-CHOP (Santa Cruz Biotechnology, sc-575, 1:1000 dilution); Goat anti-ApoB (Millipore, #AB742, 1:1000); Alexa Fluor 568 anti-Mouse Secondary Antibody (Invitrogen, #A-10037, 1:1000 dilution); Alexa Fluor 488 anti-Mouse Secondary Antibody (Invitrogen, #A-21202, 1:1000 dilution); Alexa Fluor 594 anti-Rabbit Secondary Antibody (Life Technologies, #A-11037, 1:1000 dilution); Alexa Fluor 488 anti-Rabbit Secondary Antibody (Invitrogen, #A-11008, 1:1000 dilution); Horseradish peroxidase-linked secondary antibodies were purchased from Jackson Immuno-Research Laboratories(1:5000 dilution). Lipofectamine 3000, Oligofectamine, Bodipy 493/503, and LipidTOX™ Deep Red Neutral Lipid dye were from ThermoFisher. Infinity TG Reagent were from ThermoFisher. cOmplete Mini EDTA-free tablets were from Roche Diagnostics. [9, 10-$^3$H]-OA and [2-$^3$H]-glycerol were purchased from Perkin Elmer Life sciences. [9,12-$^{14}$C]-LA was from American Radiolabeled Chemicals. Unconjugated Phosphorothioate- and Affinity Plus-modified ASOs were custom generated by Integrated DNA Technologies. A previously validated sequence (5′-TATTTTTGGTGTATCC-3′) targeting the mouse *Pnpla3* gene was selected for all PNPLA3 knockdown studies. The specificity of knockdown was demonstrated using a chemistry-matched scrambled control ASO (5′-GGCCAATACGCCGTCA-3′). Recombinant adenovirus encoding N-terminally FLAG-tagged mouse PNPLA3 under the control of a cytomegalovirus (CMV) promoter (Ad-FLAG-PNPLA3) was custom generated by Vector Biolabs. A CMV-null adenovirus (Ad-Null)] was also obtained for use in control experiments. AAV-TBG-null (107787-AAV8) and AAV-TBG-Cre (107787-AAV8, Addgene, Watertown, Ma) were prepared by the University of Pennsylvania Vector Core.

## PCR cloning of cDNA and site-directed mutagenesis

DNA plasmids encoding flag-tagged mouse G0S2 was constructed as previously reported. For construction of PNPLA3 expression plasmid, total RNA was prepared from mouse liver using the RNeasy Mini Kit (Qiagen) according to the manufacturer's instruction. cDNA was prepared from mRNA using High-Capacity cDNA Reverse Transcription Kit (ThermoFisher). The sequence containing the complete open reading frame of mouse PNPLA3 fused at the 5′ end with sequence of a FLAG or Myc tag was amplified by PCR using Phusion DNA polymerase (ThermoFisher), and then cloned into DUAL2-CCM adenoviral shuttle vector (Vector Biolabs) via BamHI/BglII and XhoI double digestion and ligation. The PNPLA3 primers designed to create restriction sites for subsequent cloning strategies are as follows: forward, 5′-GCA GAT CTG CCA CCA TGG ATT ACA AGG ATG ACG ACG ATA AGA TGT ATG ACC CAG AGC GCC-3′; reverse, 5′-GCG CGA TCG CCA TGA CCC AGA GCG CCG CT-3′. Deletion and point mutations were generated by using the QuickChange Site-Directed Mutagenesis Kit (Agilent Technologies) according to the manufacturer's guidelines.

## Cell culture

HeLa ATGL$^{-/-}$[62] and Huh7 CGI-58$^{-/-}$ cells[79] were maintained in high glucose (4.5 g/L) DMEM + 10% heat-inactivated FBS + 1x Penicillin-Streptomycin (P/S). Culture was maintained in 37 °C incubator with 5% $CO_2$. At time of experiment, cells were split and onto coverslips in 6-well plates and transfected with 1 µg DNA using Lipofectamine 3000 according to manufacturer's protocol. After 4 h of incubation, transfection/transduction media was replaced with high glucose DMEM + 10% heat-inactivated FBS +1x P/S.

## Primary hepatocyte isolation and culture

Primary mouse hepatocytes were isolated from 10–12-week-old WT male mice as previously described[80,81]. Briefly, livers were perfused and digested using collagenase, and hepatocytes were gently washed and separated using differential centrifugation. Isolated hepatocytes were plated on collagen-coated 6-well plates or coverslips and cultured in high glucose DMEM (4.5 g/L) + 10%FBS + 1x Pen/Strep. After 3 h, non-adherent cells were removed. For knockdown studies, cells were either transfected with ASO (10 µg/well) using Lipofectamine 3000 according to manufacturer's protocol. For PNPLA3 overexpression, cells were infected with Ad-FLAG-PNPLA3 ($5 \times 10^6$ PFU/ well) with 5 µg/ml polybrene in serum-free DMEM. After 4 h of incubation, transfection/transduction media was replaced with high glucose DMEM + 10% heat-inactivated FBS +1x P/S. Culture was maintained in 37 °C incubator with 5% $CO_2$.

## Immunofluorescence and confocal microscopy of Hela and Huh7 cells

Following transfection cells were treated with 250 µM of the indicated fatty acid overnight in high glucose (4.5 g/L) DMEM +1x P/S + 5% lipoprotein depleted serum (LPDS). Cell lines were FA treated in high glucose DMEM. After overnight FA loading, cells well were fixed with 4% paraformaldehyde in PBS for 15 min, permeabilized by 0.25% triton X-100 for 5 min, quenched with 100 mM glycine in PBS for 20 min, and then blocked with 2% BSA in PBS for 1 h. The cells were then exposed to primary antibody for 2 h at room temperature. Following three washes with PBS, the cells were treated for 1 h with Alexa Fluor secondary antibodies. To visualize LDs, 1 µg/ml of Bodipy 493/503 was added during the incubation with secondary antibodies. Alternatively, the cells were incubated in Deep Red LipidTOX dye at 1:5000 dilution in PBS for 15 min after the secondary antibodies. Samples were mounted on glass slides with Vector Shield mounting medium and examined under a Zeiss LSM 980 inverted confocal microscope with Airyscan 2. Acquired images were processed manually with Zen Blue and ImageJ FIJI software. LD area (2D) was quantitatively determined using the ImageJ (Fiji) Auto Local Threshold tool (Bernsen method).

## Immunofluorescence and confocal microscopy of primary mouse hepatocytes

Following Adenoviral transduction or ASO transfection, primary mouse hepatocytes were loaded with 250 µM of OA or LA overnight in low glucose DMEM (1 g/L) + 1 x P/S + 5% lipoprotein depleted serum (LPDS) to simulate the hepatic environment upon fasting. After overnight FA loading, cells well were fixed immediately or replaced with high glucose (4.5 g/L) DMEM +1 x P/S + 5% LPDS to simulate the hepatic environment upon refeeding to facilitate lipogenic transcription and endogenous PNPLA3 expression, then fixed. Following fixation, cells were stained with Bodipy 493/503, mounted, and imaged as described above.

## PNPLA3 lipase assay

PNPLA3 lipase activity was assayed using a modification of a classical lipase assay in which lipid droplets are radiolabeled in cultured cells and isolated for lipase activity detection as described by Schweiger et al.[82]. One key distinction is that PNPLA3 expressing cell lysates were not separately extracted and then mixed with isolated LDs, rather protein expression was induced in the same cells in which LDs were radiolabeled and isolated; this approach was undertaken to ensure that PNPLA3 is properly localized onto the lipid droplet surface to detect lipase activity. PNPLA3$^{-/-}$ primary mouse hepatocytes were used for this experiment to eliminate endogenous PNPLA3 lipase activity. Specifically, PNPLA3$^{fl/fl}$ mice were injected with AAV-TBG-Cre at $4 \times 10^{10}$ GC/mouse via retroorbital injection; 3 weeks after injection, primary hepatocytes were isolated as described above and hepatocytes from

one liver were split into six 10 cm dishes. After cells had attached, two plates each were infected with Ad-null, Ad-FLAG-PNPLA3 WT, or Ad-FLAG-PNPLA3 I148M ($2.5 \times 10^7$ PFU/ dish) with 5 µg/ml polybrene in serum-free DMEM. After 4 h of incubation, transfection/transduction media was replaced with high glucose DMEM + 10% heat-inactivated FBS + 1 x P/S. On the following day, one plate from each transduction group (null, PNPLA3 WT, or PNPLA3 I148M) was loaded with either 100 µM OA + 37.5 µCi $^3$H-OA or 100 µM LA + 37.5 µCi $^{14}$C-LA in addition to 20 nM insulin and 20 µM ATGListatin to augment LD accumulation. After overnight incubation cells were washed twice in ice-cold PBS and then disrupted on ice by passing 10 times through a 25 G needle in 1 ml of cell extraction buffer (0.25 M sucrose, 1 mM EDTA, 1 mM DTT, 50 µM leupeptin, 1.5 µM pepstatin). Cell lysates were normalized for total TG using colorimetric TG kit (Infinity TG) using cell extraction buffer. Equal volumes of TG normalized cell extracts were subjected to crude centrifugation at $1000 \times g$, 4 °C for 10 min to remove cell debris and total supernatant was then subjected to high-speed centrifugation at $20,000 \times g$, 4 °C for 30 min to isolate LDs. Infranatant was carefully removed using a 25 G needle; the fat layer was resuspended in 1 ml cell extraction buffer and centrifuged at $20,000 \times g$, 4 °C for 10 min to wash LDs. Infranatant was once again removed and the fat layer was resuspended in 1 ml cell extraction buffer containing 20 µM ATGListatin to inhibit ATGL-mediated lipolysis; additionally 350 µl of 20% FA-free BSA was added to establish a final concentration of 5% FA-free BSA for capture of hydrolyzed fatty acids. Final reaction mixtures were then incubated at 37 °C. On 2 h increments aliquots of reaction mixtures were transferred into 15 ml conical tubes and the reaction immediately terminated by the addition of 3.25 ml organic extraction solution (10:9:7 Methanol/chloroform/heptane) and 1 ml boric acid extraction solution (0.1 $M$ potassium carbonate, pH adjusted to 10.5 with saturated boric acid) and vortexed for 5 s. Extraction tubes were centrifuged at $1000 \times g$ for 10 min and 1 ml upper phase and 1 ml lower phase were collected into separate scintillation vials containing 10 ml scintillation cocktail and analyzed for radioactivity. Data is presented as CPM in upper phase over CPM in lower phase to normalize to reaction input.

**Pulse-chase analysis and thin layer chromatography (TLC)**
To assess the fate of exogenously acquired FAs, primary hepatocytes pulse-labeled overnight in low glucose (1 g/L) DMEM with 5% LPDS and following combination of FAs: (1) 250 µM of OA:LA (1:1) and $^3$H-OA or $^{14}$C-LA; (2) 250 µM of OA containing $^3$H-OA; and (3) 250 µM of LA containing $^{14}$C-LA. Afterwards, cells were chased in FA-free, high glucose (4.5 g/L) DMEM with 1x P/S and 5% LPDS for 24 h. Lipids were then extracted from both cells and conditioned media using 2:1 chloroform: methanol. Extracted lipids were reconstituted in 5% TritonX-100 for colorimetric detection of total intracellular and secreted TG. In parallel, extracted lipids were separated on TLC plates to identify the phospholipids (PC + PE) to TG ratio of $^3$H-OA or $^{14}$C-LA labeled lipids by scintillation counting.

To assess the rate of TG secretion, primary hepatocytes were labeled overnight in low glucose (1 g/L) DMEM with 5% LPDS and 250 µM of OA:LA (1:1), 25 uM of glycerol and 12 µCi of $^3$H-glycerol. After a 24 h chase period in FA-free, high glucose (4.5 g/L) DMEM with 5% LPDS, a 500-µl aliquot of conditioned media was measured by scintillation counting to determine TG release into conditioned media.

**Animals**
All mice were maintained on a 12 h light/dark cycle and given ad-libitum access to food and water as specified. All mice were metabolically synchronized by alternative 12 h fasting and 12 h refeeding for 3 days preceding tissue collection. All procedures were performed according to protocols approved by the Mayo Clinic Animal Care and Use Committee (IACUC-2023-00005548). Primer sequences used for genotyping can be found in Supplementary Table s1.

**Diet studies.** Mice were ad-libitum fed a corn-oil enriched Western Diet (COWD), a modified Western diet with corn oil substituted for butter (D21050712i) matched to D12079B (Research Diets, New Brunswick, NJ). See Supplementary Table s3.

**PNPLA3 overexpression.** 10-week-old female C57BL/6 J wild-type and whole-body ATGL$^{-/-}$ mice[83] were maintained on chow diet and infected with Ad-null, Ad-FLAG-PNPLA3-WT, or Ad-FLAG-PNPLA3-I148M virus via retroorbital injection at $5 \times 10^8$ PFU/mouse. Four days post-injection, mice were sacrificed following a 12 h fast.

**PNPLA3 ASO studies.** 10-week-old female wild-type C57BL/6 J mice were administered 10 mg/kg ASO [in PBS twice a week for 3 weeks via IP injection. For LXR agonist administration, both groups of mice were injected 10 mg/kg T0901317 in 1:1 m-pyrrole: ethanol (5 mg/ml) on day 19 and day 20. Following metabolic synchronization, mice were sacrificed after 6 h of refeeding on day 21. For sugar water challenge, mice were given access to 10% fructose in drinking water in addition to COWD for 3 weeks. During metabolic synchronization mice retained access to sugar water. On day 21 mice were sacrificed after 12 h of solid food fasting with access to sugar water.

**PNPLA3-LKO studies.** Pnpla3-floxed female mice (C57BL/6JGpt-Pnpla3em1Cflox/Gpt, Strain# T010917) were purchased from Gem-Pharmatech. 8-10-week-old PNPLA3$^{fl/fl}$ C57BL/6J mice were administered either AAV-TBG-null or AAV-TBG-Cre at $4 \times 10^{10}$ GC/mouse via retroorbital injection. Mice were given access to 10% fructose and 10% glucose in drinking water in addition to COWD for 6 weeks. Mice retained access to sugar water throughout metabolic synchronization. On day 42, mice were sacrificed after 12 h of solid food fasting with access to sugar water.

**PNPLA3-I148M-KI studies.** PNPLA3-I148M-KI mice (C57BL/6N) were generated as described previously[61]. 9–12-week-old male KI mice or WT littermates were given access to 10% fructose and 10% glucose in drinking water in addition to COWD for 8 weeks. Mice retained access to sugar water throughout metabolic synchronization. On day 56, mice were sacrificed after 12 h of solid food fasting with access to sugar water.

**Lipidomic analysis**
Total lipids were extracted from mouse liver tissues and analyzed using multidimensional mass spectrometry (MDMS)-based shotgun lipidomics (MDMS-SL) at the University of Texas Health Science Center at San Antonio, as previously detailed[84]. In brief, liver homogenates were assayed for total protein concentration (Pierce BCA assay, Thermo Fisher Scientific, Pittsburgh, PA, USA); an internal standard mixture was then added to homogenates and total lipids were extracted from homogenates by a modified Bligh and Dyer method[85]. Lipid extracts were then normalized to an equal volume per mg protein with methanol/chloroform (1:1 by volume), then further diluted to <50 pmol of total lipids/µl prior to infusion through a nano-electrospray ionization source device (NanoMate; Advion Bioscience Ltd., Ithaca, NY, USA) coupled to a triple-quadrupole mass spectrometer (TSQ Altis, Thermo Scientific, San Jose, CA) or Quadrupole-Orbitrap mass spectrometer (Thermo Q Exactive™, Thermo Scientific, San Jose, CA)[85]. All full and tandem MS scans were automatically acquired through use of a customized sequence subroutine operated under Xcalibur software[84]. Fatty acyl chains were estimated from neutral loss-scanning deconvolution of individual fatty acids or precursor monitoring of fatty carboxylates[86,87].

**Immunoblotting analysis**
Preparation of tissue and cell lysates and subsequent immunoblotting analysis were performed as described previously[81]. Briefly, following SDS-PAGE and transfer to nitrocellulose membranes, individual

proteins were blotted with primary antibodies at appropriate dilutions. After an overnight incubation, peroxide-conjugated secondary antibodies were incubated with the membrane at a dilution of 1:5000. The signals were then visualized by chemiluminescence (Supersignal ECL, Pierce).

### RNA extraction and quantitative real-time PCR analysis
Total RNA was extracted from cells and tissue using PureLink RNA Mini Kit (Thermo Fisher Scientific, 12183025). Complementary DNA (cDNA) was synthesized from 1 μg of total RNA using High-Capacity cDNA Reverse Transcription Kit (ThermoFisher, 4368813). Quantitative real-time PCR was performed using Itaq Universal SYBR green master mix (Bio-Rad, 1725124) on a CFX96 Touch Real-time PCR machine (Bio-Rad) or QuantStudio VII instrument (Life Technologies). Each sample was tested in duplicate or triplicate. To determine the fold-change in gene expression compared to a control group, ΔΔCt was calculated. To determine the relative expression of corresponding genes, ΔCt was determined. Primer sequences used for Real-time PCR can be found in Supplementary Table s2.

### H&E staining of liver tissue
Liver tissue was fixed in 10% formalin, embedded in paraffin (FFPE), and sectioned at 10 μm in the Mayo Clinic Histology Core. Sections were stained with hematoxylin and eosin (H&E). Stained slides were imaged by Motic Slide Scanner (Motic, Hong Kong).

### Lipoprotein fractionation
Plasma samples were pooled (4-5 mice/group) and fractionated at the Vanderbilt Analytical Services Core. Specifically, lipoprotein fractions were separated from 100 µl of plasma by gel filtration column chromatography. 65 fractions were collected and the amount of TG and cholesterol in each fraction was determined using microtiter plate, enzyme-based assays at the Vanderbilt core facility.

### Isolation and EM imaging of VLDL particles
For VLDL isolation, plasma samples from mice were pooled ($n$ = 3-4/group) to a volume of 200 µl and overlaid with 300 µl of a 1.006 g/ml KBr solution in a 7 x 20 mm thick-walled polycarbonate tube. Samples were ultracentrifuged at 100,000 rpm at 16 °C for 2 h (rotor: Beckman TLA-100; Centrifuge: Beckman Optima TLX). 100 ul from the top of the tube was collected as VLDL fraction. 5 µl of VLDL preparation was applied to a carbon-coated copper grid and stained with uranyl acetate (2.0%) for 15 min. Grids were visualized with a JEOL 1400 + transmission electron microscope (TEM).

### In vivo hepatic triglyceride secretion assay
After 3 days of metabolic synchronization, mice were intraperitoneally injected with 10 mg/kg Poloxamer-407. Blood samples were collected in plasma separation tubes serially every hour via cheek bleeding. Samples were centrifuged at 12,000 rpm and measured for TG concentration via colorimetric TG kit (Infinity TG).

### Statistics and reproducibility
All graphs and statistical analyses were generated using GraphPad Prism 9. Bar graphs represent Mean + S.D. Violin plots represent median and upper and lower quartile values. p values were determined by two-tailed student's $t$-test using GraphPad Prism 9. Statistical details can be found within the figures, with legends for p values indicated in figure legends. Where deemed necessary, ns was included to emphasize no statistical significance; where no value is given, no significance was present. Microscopic and immunoblotting data shown were derived from representative experiments that were repeated three times or more. The experiments were not randomized and the investigators were not blinded to allocation during experimental and outcome assessment.

### Reporting summary
Further information on research design is available in the Nature Portfolio Reporting Summary linked to this article.

## Data availability
All datasets generated and analyzed in this study are provided within the manuscript and the Supplementary Information file. Raw lipidomics data are provided in the accompanying files. Source data are provided with this paper.

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

## Acknowledgements

We thank Dr. Liqing Yu at University of Maryland for kindly sharing CGI-58$^{-/-}$ Huh7 cells. We thank Dr. Rudolph Zechner at University of Graz for kindly sharing access to ATGL$^{-/-}$ mice. We thank the Vanderbilt Analytical Services Core supported by NIH grant DK020593 (DRTC) for performing FPLC analysis. We thank the Mayo Microscopy and Cell Analysis Core for experimental and technical support on TEM imaging of lipoproteins. We thank the Mayo Clinic Histology Core for help with liver sectioning and staining. This project was supported by research grants from the National Institutes of Health (DK089178 and DK109096) to J.L.), This work was partially supported by National Institutes of Health training grant DK124190 to S.J. S.J. was also supported financially by the Mayo Clinic Graduate School of Biomedical Sciences.

## Author contributions

Conceptualization: S.J., J.L.; Experimental Design: S.J.; Funding acquisition: S.J., X.H., J.L.; Investigation: S.J., H.B., A.A., C.M., Y.C., S.B.; Formal Analysis: S.J., H.B.; Resources: D.L., K. M-B., X.H., J.L.; Supervision: X.H. and J.L.; Original Draft Writing: S.J., J.L; Final Draft review and editing: All authors.

## Competing interests

KM-B and DL are employed by and have shares in AstraZeneca. The remaining authors declare no competing interests.
