## [Peer Review File · Nature Communications]

PNPLA3 is a triglyceride lipase that mobilizes polyunsaturated fatty acids to facilitate hepatic secretion of large-sized very low-density lipoproteinREVIEWER COMMENTS

Reviewer #1 (Remarks to the Author):

In this study, the authors identified a link between WT PNPLA3 and very-low-density lipoprotein (VLDL) secretion with mechanisms involving polyunsaturated fatty acid (PUFA) transfer from triglycerides to phospholipids to facilitate VLDL lipidation. Several models and techniques including AAV- or ASO-mediated deficiency of PNPLA3, custom Western diets enriched in PUFAs, and in vitro and in vivo metabolic tracing were implemented to highlight reduced hepatic PUFA flux to phospholipids due to ATGL-independent impaired lipolysis and subsequently reduced lipid secretion via VLDL under PNPLA3 deficiency. These results elucidate PNPLA3 as a critical mediator of hepatic VLDL secretion and identify impaired VLDL secretion as a potential mechanism for the susceptibility to developing liver steatosis and NAFLD in patients carrying the I148M variant.

The study is executed well and characterizes distinct functions of WT PNPLA3 both in vitro and in vivo, which in turn can elucidate novel mechanisms contributing to the pathological defects of the I148M variant. We recommend minor revisions with detailed analyses of specific cohorts to further support their hypotheses and provide insight on phospholipid remodeling and secretion in the in vivo models.

Major points:

1. Figure 3: What happens to circulating and/or hepatic phospholipids in PNPLA3 AAV liver KO mice challenged with fructose water? Is there any change in fatty acid profile to decrease PUFA-containing phospholipids? Correspondingly, is the triglyceride fatty acid profile enriched in PUFAs that were not hydrolyzed? Though these changes are highlighted in the next figure in a different model, it would be beneficial to know the results in this model as well.
2. Figure 5: Does the hepatic triglyceride profile match the profile of VLDL secreted triglycerides with PNPLA3 ASO? Or is it a different composition of triglycerides being preferentially secreted? Similarly, is there a difference in secreted phospholipid abundance? More specifically, does the secreted phospholipid fatty acid profile match the hepatic phospholipid composition? This poloxamer 407 model is valuable to highlight the secreted versus hepatic fatty acid composition of these lipid classes, especially phospholipids, in vivo as the following figure highlighting similar points is in vitro.
3. Figure 7: The results in fatty acid composition show an increase of palmitate in triglycerides and a reduction of palmitate in PE and PC in addition to changes in PUFAs. Therefore, is the PNPLA3 I148M lipase activity not specific to PUFAs? Previous results in Figure 4F-G demonstrate WT PNPLA3 ASO reduced PUFA composition with no changes to saturated fatty acids, unlike the PNPLA3 I148M results in this figure. Presenting the phospholipid fatty acid composition as percent composition may help illustrate this point as mentioned in minor point 5.

Minor points:

1. Figure 3K: Please include statistical significance in plot.
2. Figure 6I: The title and text imply the y axis should be PC:TC ratio but the y-axis label is TG:PC ratio. Which one is correct?
3. Figure 7C: The title says total FA abundance but y axis says percent composition, just like 7D. Which one is correct?
4. Figure 7G-H: What is the percent composition of fatty acids in PC and PE? Triglycerides are presented in percent composition, but phospholipids are only described in nmol/mg protein. In order to make a better comparison of the transfer of fatty acids, it would be appropriate to present both in percent composition to really understand the enrichment/compositional change as reduction in abundance does not imply compositional change.

Reviewer #2 (Remarks to the Author):

The manuscript entitled "Substrate-Specific Function of PNPLA3 Facilitates Hepatic VLDL-Triglyceride Secretion During Stimulated Lipogenesis" by Johnson et al., describes a role for a

mutant form on the putative lipase PNPLA3 i.e. the I148M mutant, in impaired hepatic triglyceride (TG) secretion, and tries to provide a biochemical explanation for how this mutation causes hepatic steatosis in humans. While histological and pathological data seems to suggest this, the biochemical and lipidomics evidence provided in this manuscript are not convincing enough to support this claim, and as a result, this interesting study falls short of the claims put forth in this manuscript.

The major concerns in view of this reviewer are:

1. All the lipidomics measurements described in this paper are adapted from Yang et al., *Anal Chem*, 2009. 81(11): p. 4356-68. Firstly, this study is now over 14 years ago, and these methods are no longer used in the field for quantitative lipid measurements. Secondly, based on the information provided by the authors, i.e. the write up in the methods, and this reference, there is no way the authors can accurately measure the individual fatty acid content in various lipid classes that they describe in the manuscript. Such measurements can only be done using well established MRM methods, which the authors haven't seemed to use. For example: if a diacylated lipid (e.g. PC, PE or DAG) has a lipid composition of say 36:2, it can have several possibilities: 18:0/18:2, 18:1/18:1, 16:1/20:1, 16:0/20:2, 14:0, 22:2 etc Based on the method used by the authors, how can they distinguish between these species for the same mass. The authors need to re-visit this analysis to ensure any validity in their claims.

2. Throughout the manuscript, the authors over-stress on the PUFA-specific hydrolysis activity of PNPLA3. If this is the case, the author need to show via direct biochemical (lipase) assays that PNPLA3 indeed has this substrate preference, and only then can this claim be justified. There is no direct biochemical assay in support of this throughout the manuscript.

3. Based on the fatty acid data, the effects on 16:0 and 18:1 containing lipids (which are the most abundant) seemed very pronounced. Recent studies from Farese and walther lab has shown that alterations in lipid droplet associated lipases tend to activate or repress activity of other lipases, that act together in regulating lipid levels in cells and tissues. The lipidomics data provided by the authors could suggest that PNPLA3 acts majorly on abundant 16:0 or 18:1 fatty acid lipids, and to compensate for these changes the cell/tissue activates/represses other lipases to act on PUFAs. How can the authors exclude this possibility?

4. There are now several mechanisms worked out for hepatic TG secretion (see work from Roop Mallik) under various physiological conditions. It is not clear how mutations to PNPLA3 fits into these models? This is particularly important, because all the effects described in the manuscript by the authors are seen only in the absence of ATGL, so under what specific physiological conditions would PNPLA3 fit into this model?

5. Finally, biochemically how does I148M mutations alter the activity of PNPLA3? Is the localisation affected or is this activity affected? What is the basis for loss of function? Can this mutation be mapped onto a structure of PNAPL3 to explain this?

Response to Review Comments

We thank the editor and the reviewers for their most helpful comments and suggestions. We have addressed each specific point and modified the manuscript accordingly.

To Reviewer 1:

Major points:

1. Figure 3: What happens to circulating and/or hepatic phospholipids in PNPLA3 AAV liver KO mice challenged with fructose water? Is there any change in fatty acid profile to decrease PUFA-containing phospholipids? Correspondingly, is the triglyceride fatty acid profile enriched in PUFAs that were not hydrolyzed? Though these changes are highlighted in the next figure in a different model, it would be beneficial to know the results in this model as well.

We conducted lipidomics analysis on plasma and liver samples obtained from AAV PNPLA3-LKO mice. As shown below, the new data did not uncover any compositional differences in circulating or hepatic TGs and phospholipids.

Lipidomic profile of PNPLA3 L-KO mice with access to sugar in the drinking water. (A-J) 8–10-week-old PNPLA3^{fl/fl} mice injected with either AAV-TBG-Null or AAV-TBG-Cre were fed COWD and given access to fructose/glucose in drinking water for 6 weeks (n=8-10/group). Shotgun lipidomics was performed on whole liver tissue (A-F, n=5/group) and plasma VLDL (G-J, n=4/group). Individual FA species composition of liver TG, PC, and PE presented as absolute abundance (A-C) and as percentage of the total FA pool (D-F). Individual FA species composition of VLDL TG and PC presented as absolute abundance (G-H) and as percentage of the total FA pool (I-J). (G) Comparison of lipogenic stimulus strength between PNPLA3 models by liver mRNA expression of PNPLA3 and SREBP-1c. Comparison groups were (1) untreated WT mice, (2) mice on glucose/fructose drinking water for 6 weeks (PNPLA3^{fl/fl} + AAV-Null), and (3) mice treated with LXR agonist T09 (Control ASO). *p<0.05, **p<0.01, ***p<0.001 ****p<0.0001, t test.

Regarding the observed discrepancy in the changes of FA profiles between our ASO-mediated knockdown and the AAV-mediated liver-specific knockout mouse models, we have added the following assessment in the Discussion:

“A retrospective examination of the lipidomic profile in PNPLA3 LKO mice with access to glucose/fructose in their drinking water revealed no significant changes in PUFA enrichment in either TGs or phospholipids (data not shown). It is plausible that acute (LXR agonism) and chronic (sugar-infused drinking water) lipogenic stimulations might differentially influence the mechanism of PNPLA3-mediated hepatic TG secretion. Under chronic stimulation, the negative impact of PNPLA3 loss on VLDL may be relatively more subtle, yet it could lead to hepatic steatosis over time. Additionally, in LKO mice, adipose-derived PUFAs may contribute to hepatic phospholipid desaturation after a 12-hour solid food fast, resulting in less pronounced differences in hepatic FA compositions at both bulk tissue and plasma levels. Conversely, our knockdown approach involved unconjugated ASO, which may effectively target PNPLA3 in the liver as well as in adipose tissue.”

2. Figure 5: Does the hepatic triglyceride profile match the profile of VLDL secreted triglycerides with PNPLA3 ASO? Or is it a different composition of triglycerides being preferentially secreted? Similarly, is there a difference in secreted phospholipid abundance? More specifically, does the secreted phospholipid fatty acid profile match the hepatic phospholipid composition? This poloxamer 407 model is valuable to highlight the secreted versus hepatic fatty acid composition of these lipid classes, especially phospholipids, in vivo as the following figure highlighting similar points is in vitro.

We have included information on the lipidomic profile of secreted lipids in Figure 5. In the Discussion, we have also included comments on the observed discrepancy in compositional changes between the liver and the VLDL lipids.

“In vivo, our lipidomic data indicate a significant role of PNPLA3 in the intrahepatic mobilization of PUFAs from TGs to phosphatidylcholine. This effect is accompanied by a reduction in the total incorporation of PUFAs into both VLDL-TGs and VLDL-phospholipids. The absence of compositional changes in VLDL lipids suggests that the desaturation of intracellular membranes mediated by PNPLA3 facilitates the overall VLDL lipidation rather than the incorporation of specific PL species onto VLDL. The absence of a known mechanism for selectively transferring PUFA-phospholipids onto VLDL along with the established impact of membrane desaturation on MTTP activity further support this interpretation. Interestingly, poorly lipidated VLDL lacking differences in PUFA-PC composition has also been observed in human carriers of the steatosis-associated TM6SF2 E167K variant.

3. Figure 7: The results in fatty acid composition show an increase of palmitate in

triglycerides and a reduction of palmitate in PE and PC in addition to changes in PUFAs. Therefore, is the PNPLA3 I148M lipase activity not specific to PUFAs? Previous results in Figure 4F-G demonstrate WT PNPLA3 ASO reduced PUFA composition with no changes to saturated fatty acids, unlike the PNPLA3 I148M results in this figure. Presenting the phospholipid fatty acid composition as percent composition may help illustrate this point as mentioned in minor point 5.

We thank the reviewers for pointing out that the initial presentation of our phospholipid FA composition data in Figure 7 may suggest direct effects of PNPLA3 on 16:0 mobilization. In response, we have included additional representations of this data as percent composition in panels I and J. When depicted as percent abundance, 16:0 is indeed unaffected with overexpression of PNPLA3 I148M.

We also emphasize a key distinction between the experimental approaches in our overexpression and knockdown models, particularly regarding the nutritional conditions under which tissues were collected. As described in figure legends and Materials and Methods, tissues were collected after a 12-hour fast in Figure 7, whereas mice were refed to induce lipogenesis in Figure 4. This difference may explain the observed total enrichment of palmitate with PNPLA3 I148M overexpression. Notably, previous evidence suggests that the PNPLA3 I148M variant substantially inhibits ATGL, which is primarily expressed in the liver during fasting and is known to hydrolyze all TG substrates. On the other hand, our data demonstrate PNPLA3 as a lipase that specifically mobilizes PUFA-containing TGs in lipogenic settings. To further illustrate the substrate specificity of PNPLA3, we have included new data from a biochemical lipase assay as Figure 1I to demonstrate the preference of PNPLA3 for PUFA-TG substrates *in vitro*.

Minor points:

1. Figure 3K: Please include statistical significance in plot.

Indication of statistical significance has been included.

2. Figure 6I: The title and text imply the y axis should be PC:TC ratio but the y-axis label is TG:PC ratio. Which one is correct?

Presentation of this data as TG:PC ratio was incorrect and has been changed to PC:TG ratio.

3. Figure 7C: The title says total FA abundance but y axis says percent composition, just like 7D. Which one is correct?

Labeling of the y axis as percent composition was incorrect and has been changed to nmol/mg protein as a reflection of the total FA abundance.

4. Figure 7G-H: What is the percent composition of fatty acids in PC and PE? Triglycerides are presented in percent composition, but phospholipids are only described in nmol/mg protein. In order to make a better comparison of the transfer of fatty acids, it would be appropriate to present both in percent composition to really understand the enrichment/compositional change as reduction in abundance does not imply compositional change.

Data indicating the percent composition of FA in PC and PE have been added, and further insight is included in response to Major Point #3.

To Reviewer 2:

Major concerns:

1. All the lipidomics measurements described in this paper are adapted from Yang et al., *Anal Chem*, 2009. 81(11): p. 4356-68. Firstly, this study is now over 14 years ago, and these methods are no longer used in the field for quantitative lipid measurements. Secondly, based on the information provided by the authors, i.e. the write up in the methods, and this reference, there is no way the authors can accurately measure the individual fatty acid content in various lipid classes that they describe in the manuscript. Such measurements can only be done using well established MRM methods, which the authors haven't seemed to use. For example: if a diacylated lipid (e.g. PC, PE or DAG) has a lipid composition of say 36:2, it can have several possibilities: 18:0/18:2, 18:1/18:1, 16:1/20:1, 16:0/20:2, 14:0, 22:2 etc Based on the method used by the authors, how can they distinguish between these species for the same mass. The authors need to re-visit this analysis to ensure any validity in their claims.

We thank the reviewer for pointing out the lack of details provided in our Materials and Methods section. We have addressed these concerns by including more specifics regarding the shotgun lipidomics methodology used for this study.

We believe that our oversight may have led the reviewer to misunderstand our multi-dimensional mass spectrometry-based shotgun lipidomics (MDMS-SL) technological platform. The paper we cited by Yang et al. in *Anal Chem* only provides the principles of the MDMS-SL platform. However, as the reviewer rightly pointed out, the applications of this platform have been advanced greatly since 2009. For example, we developed and published methodology to advance the platform for quantitative measurement of all lipids containing carboxylic acid in 2013 (**PMID: 23971716**), diacylglycerol in 2014 (**PMID: 24432906**), all lysophospholipids in 2015 (**PMID: 25860968**), polyphosphoinositides in 2016 (**PMID: 28193056**), FAHFA in 2020 (**PMID: 32138907**), monohexosyl DAG in 2020 (**PMID: 32891384**), and many more. Many of these papers are published in *Analytical Chemistry* based on the principles described in the 2009 paper. By using the MDMS-SL approach, over 100 papers have been published by the Han lab and their collaborators in just the last five years; as such, this platform represents a well-recognized, advanced, comprehensive, and accurate approach for lipidomics research (see recently published review articles **PMID: 32647401**, **PMID: 34953866**).

Regarding the assessment of fatty acyl constituents, our approach used either the neutral loss of corresponding fatty acid and fatty ketenes, or precursor monitoring of fatty carboxylates as outlined in Table 2 of Yang paper (**PMID: 19408941**). Applying these neutral loss or precursor scans corresponding to the aliphatic moieties enables us not only to identify these fatty acyl chains, but also quantify them as well as assign their regiospecificity. In fact, neutral loss and/or precursor ion scans are comparable to MRM (see extensive discussion in the review article **PMID: 21755525**).

As aforementioned, by using multiple scans corresponding to all the potential fatty acyl chains, the identities and regioselectivity can be identified. Please see **PMID: 24432906** for detailed description for analysis of diacylglycerol species.

2. Throughout the manuscript, the authors over-stress on the PUFA-specific hydrolysis activity of PNPLA3. If this is the case, the author need to show via direct biochemical (lipase) assays that PNPLA3 indeed has this substrate preference, and only then can this claim be justified. There is no direct biochemical assay in support of this throughout the manuscript.

Our experiments using both cultured cells and mice demonstrate that PNPLA3 specifically degrades PUFA-induced lipid droplets (LDs) in a CGI-58-dependent but ATGL-independent manner. The finding that mutation of the catalytic serine 47 to alanine (S47A) almost completely abolished PNPLA3's ability to degrade linoleic acid-induced LDs in cells further suggests a hydrolase activity specifically toward PUFA-TGs. However, we acknowledge that the absence of a direct biochemical assay demonstrating PNPLA3 lipase activity was a weakness in our initial submission. Despite numerous prior attempts, we were unable to detect TG hydrolase activity using an established in vitro assay wherein radiolabeled TG substrates are emulsified by phosphatidylcholine from egg yolk and phosphatidylinositol from soybean (**PMID: 24529439**).

In light of this reviewer's comment and upon reflection on our previous experimental conditions, we hypothesized that our inability to capture PNPLA3 lipase activity in vitro is likely due to the inability of extracted PNPLA3 to properly localize to the surface of emulsified TG substrates. In this regard, PNPLA3 is not known to harbor the lipid-anchoring domain of ATGL, and third-party partners may be needed to assist its LD localization in cells. Given the crucial role of surface phospholipids in allowing lipase access to the TG core in LDs, it is also possible that the composition of phospholipids used to emulsify TG substrates in vitro may not resemble that of the phospholipid surface of LDs in vivo. On these bases, we modified our experimental approach and utilized primary hepatocytes to concurrently express PNPLA3 and produce labeled TG-LDs, ensuring association of the enzyme and LDs with the proper phospholipid composition prior to LD extraction. By adopting this approach, we were finally able to capture significant FA release mediated by PNPLA3 pre-localized to TG-LDs. Our new data demonstrate the substrate preference of PNPLA3 for linoleic acid-containing over oleic acid-containing LDs (Figure 1I). That the I148M mutant is significantly impaired in its hydrolytic activity in this setting indicates its nature as a loss-of-function variant.

3. Based on the fatty acid data, the effects on 16:0 and 18:1 containing lipids (which are the most abundant) seemed very pronounced. Recent studies from Farese and Walther lab has shown that alterations in lipid droplet associated lipases tend to activate or repress activity of other lipases, that act together in regulating lipid levels in cells and tissues. The lipidomics data provided by the authors could suggest that PNPLA3 acts majorly on abundant 16:0 or 18:1 fatty acid lipids, and to compensate for these changes the cell/tissue activates/represses other lipases to act on PUFAs. How can the authors exclude this possibility?

We do not contend that PNPLA3 may interact with other lipases in the regulation of lipid homeostasis. ATGL is considered the rate-limiting TG lipase in the liver during fasting. It has

been demonstrated that the I148M mutant of PNPLA3 competes with ATGL for its cofactor CGI-58, resulting in the inhibition of ATGL-mediated lipolysis (**PMID: 24917523**). The increased percent enrichment of 16:0 in TG with the expression of PNPLA3^{I148M} during fasting, as shown in Figure 7D, is conceivably due to impeded ATGL-mediated lipolysis.

On the other hand, we would like to emphasize that our model implicates PNPLA3 in facilitating TG secretion during lipogenic stimulation, making available stored PUFAs, which cannot be synthesized de novo. Upon PNPLA3 loss, impaired TG secretion would lead to the retention of not only PUFA-containing TGs but also TGs containing 16:0 and 18:1 FAs, as these are the primary products of de novo lipogenesis.

The data presented in Figures 1 and 2, utilizing both cell and animal models, were designed as overexpression models to investigate the biochemical activity of PNPLA3. In contrast, the data presented in Figures 3 through 6 were designed as loss-of-function studies aimed at investigating the physiological function of PNPLA3. As the reviewer rightly points out, this has consequences for many lipid species, not just PUFAs.

4. There are now several mechanisms worked out for hepatic TG secretion (see work from Roop Mallik) under various physiological conditions. It is not clear how mutations to PNPLA3 fits into these models? This is particularly important, because all the effects described in the manuscript by the authors are seen only in the absence of ATGL, so under what specific physiological conditions would PNPLA3 fit into this model?

We appreciate the reviewer's interest in understanding how our work contributes to the specific mechanisms of VLDL synthesis, maturation and secretion. In our humble opinion, the primary contribution of our present work lies in uncovering the physiological relevance of PNPLA3 as a PUFA-TG lipase. Specifically, our data demonstrate that, unlike fasting-induced ATGL, PNPLA3 is upregulated and exerts its enzyme function during lipogenesis-driven VLDL production.

Given the established importance of membrane desaturation for efficient MTP-mediated TG transfer onto VLDL particles (**PMID: 25898003**), our findings suggest a novel mechanism by which stored PUFAs are mobilized to support membrane phospholipid desaturation and VLDL production during lipogenic stimulation. It's worth noting that ATGL is intentionally absent in some of our overexpression models to minimize the confounding effects of ATGL-mediated lipolysis on our investigation into the biochemical function of PNPLA3. Despite this, the overexpression of PNPLA3 still reduces 18:2 in liver TG of wild-type mice (see Figure 2E and 2F). Also, all our loss-of-function models are conducted in mice with intact endogenous ATGL, though its expression is conceivable low during lipogenic stimulation.

5. Finally, biochemically how does I148M mutations alter the activity of PNPLA3? Is the localisation affected or is this activity affected? What is the basis for loss of function? Can this mutation be mapped onto a structure of PNAPL3 to explain this?

We believe that it has been adequately addressed in previous work. Despite the elusive nature of PNPLA3 crystallization, modeling of PNPLA3 molecular structure has indicated that isoleucine 148 folds directly adjacent to the active site residues (S47, D166) (**PMID: 20034933, PMID: 29935383**). Additionally, other studies have shown that the I148M mutation does not disrupt LD localization (**PMID: 20034933, PMID: 30802989**), as also can be appreciated from our data presented in Figure 1C.

REVIEWERS' COMMENTS

Reviewer #1 (Remarks to the Author):

the authors have addressed my concerns.

Reviewer #2 (Remarks to the Author):

The authors have satisfactorily addressed my concerns, and recommend publication of this manuscript.